# Targeting mTOR and Metabolism in Cancer: Lessons and Innovations

**DOI:** 10.3390/cells8121584

**Published:** 2019-12-06

**Authors:** Cedric Magaway, Eugene Kim, Estela Jacinto

**Affiliations:** Department of Biochemistry and Molecular Biology, Rutgers-Robert Wood Johnson Medical School, Piscataway, NJ 08854, USA; cgm95@rwjms.rutgers.edu (C.M.); eugene.jae.kim@gmail.com (E.K.)

**Keywords:** mTORC1, mTORC2, metabolism, rapalogs, mTOR inhibitors, cancer metabolism, mTOR in immunotherapy, nutrient metabolism, kinase inhibitors, mTOR signaling

## Abstract

Cancer cells support their growth and proliferation by reprogramming their metabolism in order to gain access to nutrients. Despite the heterogeneity in genetic mutations that lead to tumorigenesis, a common alteration in tumors occurs in pathways that upregulate nutrient acquisition. A central signaling pathway that controls metabolic processes is the mTOR pathway. The elucidation of the regulation and functions of mTOR can be traced to the discovery of the natural compound, rapamycin. Studies using rapamycin have unraveled the role of mTOR in the control of cell growth and metabolism. By sensing the intracellular nutrient status, mTOR orchestrates metabolic reprogramming by controlling nutrient uptake and flux through various metabolic pathways. The central role of mTOR in metabolic rewiring makes it a promising target for cancer therapy. Numerous clinical trials are ongoing to evaluate the efficacy of mTOR inhibition for cancer treatment. Rapamycin analogs have been approved to treat specific types of cancer. Since rapamycin does not fully inhibit mTOR activity, new compounds have been engineered to inhibit the catalytic activity of mTOR to more potently block its functions. Despite highly promising pre-clinical studies, early clinical trial results of these second generation mTOR inhibitors revealed increased toxicity and modest antitumor activity. The plasticity of metabolic processes and seemingly enormous capacity of malignant cells to salvage nutrients through various mechanisms make cancer therapy extremely challenging. Therefore, identifying metabolic vulnerabilities in different types of tumors would present opportunities for rational therapeutic strategies. Understanding how the different sources of nutrients are metabolized not just by the growing tumor but also by other cells from the microenvironment, in particular, immune cells, will also facilitate the design of more sophisticated and effective therapeutic regimen. In this review, we discuss the functions of mTOR in cancer metabolism that have been illuminated from pre-clinical studies. We then review key findings from clinical trials that target mTOR and the lessons we have learned from both pre-clinical and clinical studies that could provide insights on innovative therapeutic strategies, including immunotherapy to target mTOR signaling and the metabolic network in cancer.

## 1. Introduction

In 1956, Otto Warburg wrote that if we know how cancer cells have “damaged respiration and excessive fermentation”, then we understand the origin of cancer cells [1]. Warburg was referring to the abnormal metabolism of glucose in cancer. Both normal and cancer cells metabolize nutrients, primarily glucose, to produce energy in the form of ATP. There are two ways for cells to generate ATP. First, through the more evolutionarily primitive means via glycolysis, a process that converts glucose to pyruvate without the need for oxygen (anaerobic). This process is also referred to as fermentation, as lower organisms ferment glucose to ethanol, whereas higher organisms can convert pyruvate to lactate in the cytoplasm. Secondly, ATP is also produced via respiration in the mitochondria. Pyruvate from glycolysis enters the mitochondria, resulting in the production of abundant ATP in the presence of oxygen (aerobic), hence the term respiration. However, Warburg observed that cancer cells increase their fermentation, referring to the anaerobic process of glucose metabolism despite the presence of oxygen. This has been termed as aerobic glycolysis or the Warburg effect. The increased fermentation favors the conversion of pyruvate to lactate over pyruvate metabolism in the mitochondria. Warburg proposed that the “driving force of the increased fermentation is the energy deficiency” in order to generate sufficient ATP in lieu of the defective respiration process. How this happens in cancer cells was then mysterious, but more modern approaches to study cancer metabolism have now provided clues that may help us understand the origin of cancer and how we can eradicate them more effectively.

More than a decade after Warburg’s observations, an expedition in the island of Rapa Nui to collect soil bacteria for the purpose of isolating active natural compounds unearthed a drug that would later prove instrumental in understanding the anomalous metabolism of cancer cells. More importantly, this drug, named rapamycin, has served as a prototype for the development of more effective compounds to treat cancer as well as other metabolism-related disorders. Rapamycin forms a complex with the prolyl isomerase FKBP12 and together binds to mTOR, a protein kinase that plays a central role in controlling growth and metabolism [2]. Since the identification of mTOR, numerous studies have unraveled how mTOR signaling reprograms metabolism to acquire nutrients from the environment and intracellular sources in order to generate ATP as well as intermediates for macromolecule synthesis [3]. This central function of mTOR in metabolic reprogramming could be targeted using mTOR inhibitors and thus prevent tumor growth and malignancy [4]. The first generation mTOR inhibitors, rapamycin analogs (rapalogs), have shown promising results in pre-clinical studies but with modest results in the clinic [5]. Nevertheless, some of the rapalogs have now been approved by the FDA for the treatment of specific types of cancer. A limitation of rapamycin is that it is only an allosteric mTOR inhibitor and does not fully block its activity. Second generation mTOR inhibitors that target the catalytic site of mTOR to block mTOR activity have been developed [6]. Pre-clinical studies with these inhibitors have confirmed their potency in preventing cell growth and proliferation. However, results from clinical trials have been sobering due to their toxic nature. Hence, there is a need to improve therapeutic strategies in order to gain benefit from these compounds. Studies from the use of these mTOR inhibitors have also unraveled that tumors exhibit such metabolic plasticity and that other mTOR-independent mechanisms can be triggered to compensate for the block in mTOR activity, thus allowing cells to acquire nutrients and metabolize them for growth and proliferation. Here, we discuss the metabolic functions of mTOR that have emerged from pre-clinical studies using cells and animal models. We then review the results from studies utilizing different strategies of mTOR inhibition both from pre-clinical and clinical studies and discuss the lessons from the successes and failures gleaned from these studies. We mention rational therapeutic approaches to more specifically target tumors that will have high sensitivity to mTOR inhibition. Lastly, we discuss recent innovative strategies in manipulating the tumor microenvironment using mTOR inhibitors to target the growing tumor or to improve immunotherapy for cancer treatment.

## 2. mTOR Signaling and its Role in Metabolism

Early studies in yeast unraveled that TOR is involved in the regulation of growth or cell mass accumulation in response to the presence of nutrients [7,8]. Studies in other organisms such as *Drosophila* and mammals further corroborated the critical role of mTOR in promoting not only cell growth but also organismal growth [9]. The elucidation of the function of mTOR in protein synthesis and autophagy provided clues on its role in nutrient sensing and anabolic metabolism [10,11]. Genome-wide screening further uncovered the effect of rapamycin on metabolic genes, revealing that TOR/mTOR mediates the expression of genes involved in nutrient metabolism [12,13,14,15]. mTOR is part of two structurally distinct complexes, mTORC1 and mTORC2. The conserved components of mTORC1 include mTOR, raptor and mLST8 whereas mTORC2 consists of mTOR, rictor, SIN1 and mLST8 (Figure 1). Genetic studies that ablated components of the mTOR complexes in a tissue-specific manner also provided support on the role of mTOR on glucose, amino acid, lipid, nucleotide metabolism and other biosynthetic pathways [16,17,18]. In addition to promoting anabolic metabolism, mTOR also functions to negatively regulate catabolic processes such as autophagy. Altogether, these findings unraveled how mTOR controls cell growth via its central role in metabolism.

### 2.1. Signaling to mTOR

mTOR as part of mTORC1 is active in the presence of nutrients such as amino acids [19]. Several amino acid transporters, including the transporters for glutamine (SLC1A5/ASCT2) and leucine (SLC7A5/LAT1, which imports Leu in exchange for Gln efflux by SLC3A2/CD98/4F2hc), have been linked to mTORC1 activation and their overexpression is often associated with malignancies [20,21,22,23]. The activation of mTORC1 occurs via recruitment to the surface of the lysosomes, a major hub for the degradation and recycling of macromolecules. When nutrients are abundant, mTORC1 is activated via the Ras-related GTP binding proteins (Rags) [24,25]. RagA/B is bound to GTP while RagC/D is GDP-bound under amino acid sufficiency. The Rag heterodimers then interact with raptor and facilitate translocation of mTORC1 to the lysosomal surface. Amino acids such as leucine and arginine activate mTORC1 robustly via Rag-dependent mechanisms. In the case of leucine and arginine, proteins that bind to these amino acids, such as sestrin 2 and CASTOR1, respectively, mediate the activation of mTORC1 [26]. In the presence of leucine, the sestrin2/leucine interaction relieves the inhibition of GATOR1 by GATOR2, ultimately activating RagA/B and mTORC1 (Figure 1). Arginine binding to CASTOR1 also derepresses the inhibition of GATOR2 by CASTOR1, thereby activating mTORC1. Another Arg sensor, SLC38A9, an Arg gated amino acid transporter affects RagA/RagB nucleotide state and thus controls mTORC1 activation [27,28]. Methionine, on the other hand, is sensed as S-adenosylmethionine (SAM) via SAMTOR. The binding of SAM to SAMTOR disrupts the SAMTOR-GATOR1 association, thus activating mTORC1. Glutamine activates mTORC1 via Rag-dependent and -independent mechanisms [19,29]. Glutamine is metabolized during glutaminolysis to produce α-ketoglutarate. This process enhances mTORC1 activation via Rag. Glutamine also promotes mTORC1 translocation to the lysosomal surface via the ADP ribosylation factor (Arf1) GTPase, independent of Rag [30,31]. A lysosomal GPCR-like protein GPR137B also regulates Rag and mTORC1 localization to the lysosomes and plays a role in regulating the GTP loading state of RagA [32]. Leucine also activates mTORC1 via its catabolite acetyl-coenzyme A (AcCoA), independently of sestrin2 [33]. AcCoA promotes EP300-mediated acetylation of raptor at K1097, thus enhancing mTORC1 activity. This regulation of mTORC1 appears to be cell-type-specific, however. The activation of mTORC1 by amino acids is potentiated by signals from growth factors. The binding of growth factors such as insulin to receptor tyrosine kinases trigger phosphatidylinositol-3 kinase (PI3K) activation. PI3K signals are wired to mTORC1 via the tumor suppressor proteins tuberous sclerosis complex (TSC1 and TSC2), which acts to negatively regulate mTORC1 [34,35]. Upon PI3K activation, the TSC1/TSC2 complex is inactivated. TSC2 functions as a GTPase-activating protein (GAP) that inhibits the Rheb GTPase [36,37,38]. Hence, cells that lack or have inactivating mutations in TSC1 or TSC2 have increased mTORC1 activity. Rheb binds and activates mTORC1 by realignment of residues in the mTOR active site to enhance catalytic activity [37]. As discussed further below, there are a number of tumors that have upregulated mTORC1 activity due to TSC1/TSC2 inactivation. mTORC1 is also modulated by cellular energy via AMPK. AMPK modulates mTORC1 via phosphorylation of raptor and indirectly via phosphorylation of TSC2 [34,39]. Under energy-depleted conditions, AMPK becomes activated and inhibits mTORC1. Decreased oxygen also dampens mTORC1 activation via REDD1, which acts through TSC [40]. Recently, mTORC1 has been shown to also be inhibited by G-protein coupled receptor (GPCR) signaling [41]. This inhibition occurs via G_αs_ proteins, which increase cyclic adenosine 3′5′ monophosphate (cAMP), leading to the activation of PKA. PKA phosphorylates Ser791 of raptor, leading to diminished mTORC1 activity. In addition to mTORC1 activation on the lysosomal surface, mTORC1 is also activated by amino acids at the surface of the Golgi via Rab1A, another small GTPase [42]. Rab1A activates mTORC1 via promoting the interaction of mTORC1 with Rheb in the Golgi. The amino acid transporter PAT4 (SLC36A4), which is mainly associated with the Golgi interacts with mTORC1 and Rab1A [20]. Further studies are needed to determine how mTORC1 could be activated in different cellular compartments by amino acids and possibly other metabolites [43].

The activation of mTORC2 is relatively less understood than that of mTORC1 [44]. mTORC2 associates with plasma membrane, mitochondria and a subpopulation of endosomal vesicles [45]. mTORC2 is also activated by signals from growth factors. The precise mechanisms as to how mTORC2 is activated remain unclear but its component SIN1 has been shown to bind phosphatidylinositol (3,4,5)-trisphosphate PIP3 and could thus promote mTORC2 activation [46,47]. Increased receptor tyrosine kinase activation enhances PI3K activity, resulting in increased PIP3. This phospholipid attracts several signaling molecules including Akt and the 3-phosphoinositide dependent kinase 1 (PDK1). Signals from PI3K are antagonized by the lipid phosphatase tumor suppressor, PTEN. Tumors with activating mutations in PI3K or inactivating mutations or deletion of PTEN have upregulated mTORC2 signaling. PI3K activation increases phosphorylation of the mTORC2 target Akt at the allosteric site Ser473 while PDK1 phosphorylates Akt at the catalytic site at Thr308. mTORC2 also promotes the phosphorylation of PKC and SGK1 at sites homologous to the Akt-phosphorylated sites [48,49,50,51]. Additionally, mTORC2 promotes the phosphorylation of Akt and PKC at the turn motif site in a manner that occurs co-translationally and is not further enhanced by addition of growth factors [52]. Consistently with this, active mTORC2 is present in the ribosomes [53]. Interestingly, mTORC2 activation is also enhanced during nutrient limitation [54,55]. Glutamine or glucose starvation can increase phosphorylation of mTORC2 targets such as Akt. The activation of mTORC2 during glucose depletion occurs via a more direct action of AMPK on mTORC2 via phosphorylation of mTOR and possibly rictor [55]. β- and α-adrenergic signaling through GPCR also modulates mTORC2. β-adrenergic stimulation of mTORC2 signaling induces lipid catabolism in brown adipocytes and glucose uptake into skeletal muscle or brown fat cells [56]. The function of rictor in lipid catabolism in brown adipocytes is independent of Akt but involves FoxO1 deacetylation via SIRT6 [57]. α-adrenergic signaling to mTORC2 also promotes glucose uptake in cardiomyocytes [58]. Hence, unlike mTORC1 that is activated by anabolic and negatively regulated by catabolic signals, mTORC2 seems to be modulated positively by both types of signals. It is likely that mTORC2 maintains a basal level of activation, as supported by constitutive phosphorylation of some of its targets, but further elevates its activity to restore metabolic homeostasis during nutrient fluctuations.

### 2.2. Protein Synthesis

mTOR has multiple roles in the regulation of protein synthesis (Figure 2). Much of what we know about this mTOR function relates to mTORC1. mTORC1 phosphorylates two key effectors of translation, S6K1 and eIF4E-BP (4EBP). mTORC1 phosphorylates the Thr389 residue of S6K1, a serine/threonine kinase that belongs to the AGC kinase family [59]. Thr389 resides in the hydrophobic motif of S6K1, which is a motif that is common to most AGC kinases. Phosphorylation at Thr389 enables subsequent phosphorylation of S6K1 at its activation loop by PDK1. S6K1 has several downstream targets that are involved in protein synthesis, including eIF4B, a positive regulator of the 5′cap binding eIF4F complex. It also phosphorylates PDCD4, leading to its degradation and the positive regulation of eIF4B [60]. S6K1 also enhances translation efficiency of spliced mRNAs via SKAR, a component of exon-junction complexes [61]. In addition to the above functions of S6K1, it has other targets that are involved in metabolism, as discussed further below. mTORC1 also phosphorylates 4EBP at multiple sites. Phosphorylation of 4EBP promotes 5′cap-dependent mRNA translation by dissociation of 4EBP from eIF4E, thus allowing assembly of the eIF4F complex. 4EBP is also phosphorylated by other kinases to regulate its function in an mTORC1-dependent or -independent manner [62,63].

In addition to regulating key effectors of protein synthesis, mTOR also regulates synthesis of amino acids, the building blocks for protein synthesis. mTOR upregulates asparagine biosynthesis via enhancing expression of the asparagine synthetase (ASNS) in colorectal cancer cells with KRAS mutations [64]. In non-small-cell lung cancer expressing oncogenic KRAS, the increased ASNS expression occurs via Akt- and Nrf2-mediated induction of ATF4. Tumor growth is prevented by inhibition of Akt together with depletion of extracellular asparagine [65]. mTOR also regulates the amounts of amino acid transporters. Genomic studies identified neutral amino acid transporters to be decreased upon rapamycin treatment [13]. mTORC2 also plays a role in regulating amino acid transporters. mTORC2 phosphorylates Ser26 of the cystine-glutamate anti-porter xCT, thereby preventing the efflux of glutamate and influx of cystine [66]. When mTORC2 activity was disrupted genetically or pharmacologically, glutamate secretion, cystine uptake and incorporation into glutathione were enhanced. This function of mTORC2 could allow highly proliferating cancer cells to utilize glutamate for TCA anaplerosis. When nutrients become limiting or when mTORC2 signals are dampened, the influx of cystine, at the expense of glutamate efflux, would allow tumor cells to relieve redox stress.

mTOR also controls the biogenesis of ribosomes, the machinery that drives protein synthesis. mTOR positively regulates several processes involved in ribosome biogenesis, including rRNA transcription as well as the synthesis of ribosomal proteins and components of ribosome assembly [67]. The mTORC2 component rictor is recruited to the nucleolar compartment during epithelial-to-mesenchymal transition (EMT)-associated ribosome biogenesis [68]. The role of rictor or mTORC2 in this compartment requires further investigation. Since previous studies demonstrated that active mTORC2 associates with ribosomes and that it could phosphorylate nascent peptides [52,53], mTORC2 could thus have multiple ribosome-associated functions. Many tumors have high rates of ribosome biogenesis to support the augmented protein synthesis necessary for growth and proliferation. Inhibition of RNA polymerase I has been used in pre-clinical studies and is also undergoing clinical trials for cancer therapy [69]. Recently, the inhibitor of Pol I-mediated rDNA transcription (CX-5461) was combined with everolimus and was shown to synergistically increase the survival of mice with a Myc-driven lymphoma [70].

### 2.3. Glucose Metabolism

Cancer cells enhance their rate of glucose uptake and produce pyruvate at a higher rate than can be metabolized by the mitochondria. Under this condition, the excess pyruvate is diverted from being metabolized in the mitochondria to being converted to lactate in the cytosol. This glycolytic switch can occur under aerobic conditions. Genetic mutations that lead to enhanced growth factor-PI3K/Akt/mTOR signaling can drive and/or maintain this switch. Indeed, multiple points along the glycolytic pathway are influenced by mTOR via regulation of critical transcription factors such as HIF1α and Myc (Figure 3). HIF1 (hypoxia inducible factor 1) is a heterodimer consisting of an O_2_-regulated HIF1α subunit and a constitutively expressed HIF1β subunit [71]. The expression of HIF1α is dependent on mTORC1 and mTORC2 [72]. Increased HIF1α expression is sufficient to induce expression of genes whose products increase glycolytic flux [73]. While HIF1α expression is normally elevated under hypoxia, it becomes abnormally upregulated in cancer despite aerobic conditions. HIF1 expression is upregulated in many primary and metastatic human tumors [71]. Early studies in prostate cancer cells have shown that inhibiting mTOR by rapamycin blocks the growth factor- and mitogen-induced HIF1α expression [74]. Rapamycin also decreases HIF1α stabilization and transcriptional activity under hypoxic conditions [75]. mTORC1 promotes HIF1α transcription through Foxk1 [76]. By mTORC1-mediated inhibition of GSK3, Foxk1 phosphorylation is repressed and allows its accumulation in the nucleus to induce HIF1α transcription. Elevated mTORC1 activation that occurs in TSC2^−/−^ cells also increases translation of HIF1α mRNA [77] while rapamycin decreases its mRNA levels [78,79]. The control of HIF1α translation involves the mTORC1 target, 4E-BP1 [14]. Thus, mTORC1 can regulate HIF1α expression at the level of transcription and translation.

mTORC1 also modulates expression of the glycolytic genes via HIF1α. Transcriptional profiling of rapamycin-treated lymphocytes revealed altered glycolytic gene expression in these cells [13,80]. In prostate epithelial cells of transgenic mice expressing active Akt, rapamycin diminishes the levels of glycolytic enzyme genes controlled by HIF1α [81]. A combination of genomics, metabolomics and bioinformatics approaches further confirmed the involvement of mTORC1 in inducing a HIF1α-dependent transcriptional program to promote glycolysis [14]. Among the HIF1α-regulated genes that are transcriptionally upregulated in an mTORC1-dependent manner are glycolytic enzymes and VEGF. The expression of the rate-limiting glycolytic enzyme pyruvate kinase M2 (PKM2), which is exclusively expressed in proliferating and tumor cells, is also regulated transcriptionally by mTORC1 via HIF1α [82].

mTORC1 also controls glucose uptake via regulation of gene expression or membrane trafficking of glucose transporters. Cancer cells upregulate their consumption of glucose. This characteristic of cancer cells is exploited in fluorodeoxyglucose -positron emission tomography (FDG-PET) scan that is widely used to detect tumors. Rapamycin treatment in vivo reduces FDG uptake in kidney cancers with loss of the tumor suppressor von Hippel Landau (VHL1), supporting a role for mTORC1 in glucose uptake [79]. The sensitivity to mTOR inhibition is attributed to a block in translation of mRNA encoding HIF1α, a target of VHL. Glucose uptake, as well as the expression of glycolysis-associated genes and glucose metabolism-regulating miRNAs, were decreased by PI3K/mTOR inhibitors in lymphoma cells [83]. However, in another study using liver-specific *Tsc1* mutant mice, increased mTORC1 activation is accompanied by reduced glucose uptake [84]. This is likely due to the mTORC1/S6K1-mediated negative feedback loop that downregulates PI3K/Akt pathway, which plays a role in glucose transport [85]. The reduced glucose uptake under elevated mTORC1 activity is also in line with the findings that TSC-deficient cells are hypersensitive to glucose withdrawal [34]. It is also likely that the increased mTORC1 activation could upregulate the export or synthesis of alternative carbon or energy sources such as glutamine.

Another effector of mTORC1 that promotes glycolytic gene expression is the transcription factor Myc [86,87]. Myc stimulates transcription of genes involved in metabolism, ribosome biogenesis and mitochondrial function [88]. Using a bioinformatics approach, cis-regulatory elements among rapamycin-sensitive genes were shown to be regulated by Myc [14]. HIF1 and Myc have overlapping metabolic gene targets. One example of a common target gene is that encoding lactate dehydrogenase (LDH) [89,90]. LDH converts pyruvate to lactate and is a tetrameric enzyme composed of a combination of the subunits LDHA and LDHB. Rapamycin treatment of prostate cancer cell lines downregulates LDHA expression among other metabolic effectors [91]. On the other hand, the expression of LDHB is upregulated in an mTOR-dependent manner in murine embryonic fibroblasts that have deficiency in TSC1, TSC2 or PTEN and with activated Akt. The enhanced LDHB levels are critical for hyperactive mTOR-mediated tumorigenesis [92]. Another common target of HIF1α and Myc is PKM2. Unlike transcriptional activation of the PKM2 gene by HIF1α, Myc appears to regulate PKM2 expression in an mTORC1-dependent manner via the alternative splicing repressors, hnRNPs [82]. How mTORC1 can regulate Myc remains to be characterized.

Downstream mTORC1 substrates also mediate glycolytic metabolism. Knockdown of S6K1 in PTEN-deficient cells decreases HIF1α expression and glycolysis [93]. In these studies, targeting S6K1 in PTEN-deficient mouse model of leukemia delays leukemogenesis. Additionally, pharmacological or genetic inhibition of another mTORC1 target, 4E-BP1, is also sufficient to block Myc-driven tumorigenesis [94]. Studies using knockouts of negative regulators of mTORC1 further reveal how enhanced mTORC1 activation reprogram metabolism. Knockout of the GATOR1 component, NPRL2, in skeletal muscle increased pyruvate conversion to lactate while reducing its entry into the TCA cycle [95]. In turn, there was a compensatory increase in anaplerotic reactions accompanied by decreased amounts of amino acids such as aspartate and glutamine that are likely utilized for anaplerosis.

Although Warburg thought that respiration is defective in cancer cells, recent studies demonstrated that cancer cells also upregulate processes in the mitochondria. Mitochondrial oxidative phosphorylation (OXPHOS) via the respiratory chain is required by cancer cells for their proliferation. The mitochondrial import protein coiled-coil helix coiled-coil helix domain-containing protein (CHCHD4), which controls respiratory chain complex activity and oxygen consumption, promotes mTORC1 signaling and drives tumor cell growth [96]. mTORC1 also increases the expression of nucleus-encoded mitochondrial proteins via inhibition of 4EBP [97].

mTORC2 also plays a role in glycolysis. Early studies uncovered the role of Akt, an mTORC2 substrate, in the regulation of glycolytic enzymes such as hexokinase [98], PFK2 [99] as well as the glucose transporter GLUT1 [100,101]. In an experimental leukemia model, Akt activation was sufficient to increase the rate of glucose metabolism [102]. Furthermore, glioblastoma cells expressing constitutively active Akt have high rates of aerobic glycolysis. In prostate epithelial cells, activated Akt induces glycolytic genes via HIF1α [81]. On the other hand, Akt deficiency is sufficient to suppress tumor development in PTEN+/− mice [103]. Thus, Akt plays a crucial role in enhanced aerobic glycolysis. Studies that abrogated the mTORC2 component, rictor, highlighted the role of mTORC2 in glycolysis. For example, in a liver-specific knockout of rictor, glycolysis was impaired and the activity of glucokinase was reduced. Expression of a constitutively active Akt or glucokinase rescues glucose flux in these mice [104]. mTOR, most likely via Akt, also modulates pyruvate dehydrogenase (PDH), the gatekeeper enzyme of mitochondrial respiration [105]. Inhibiting PI3K/mTOR or Akt increased the phosphorylation of the E1a subunit of the PDH complex on Ser293, thus inhibiting its activity and reducing the oxygen consumption rate of head and neck squamous carcinoma cells. Akt may also function in glycolysis independent of mTOR. In glioblastoma cells, Akt1 promoted HIF1α translation in a manner that is insensitive to rapamycin or mTOR depletion [106].

mTORC2 also controls glycolysis via Akt-independent mechanisms. In glioblastoma mTORC2 promotes inactivating phosphorylation of Class IIa histone deacetylases, which then controls acetylation of FoxO1 and FoxO3 [107]. This in turn promotes upregulation of Myc. Furthermore, mTORC2 promotes acetylation of the histone H3K56 in glioma, which influences glycolytic gene expression due to enhanced recruitment of Sirt6 in the promoter of these genes [108].

Inhibiting mTORC2 activity such as by decreasing expression of its component, rictor, decreases glycolytic metabolism in cancers. In glioblastoma wherein mTORC2 activity is elevated, the expression of the large intergenic non-coding (Linc) RNA-RoR was attenuated. Increasing the expression of LincRNA-RoR led to decreased rictor expression, reduced mTORC2 activity, diminished expression of glycolytic effectors and diminished tumor growth [109].

### 2.4. Lipid Metabolism

Cancer cells undergo increased de novo lipid synthesis. Production of fatty acids and cholesterol are enhanced for biosynthesis of membranes and signaling molecules. Cell membrane lipids including phospholipids, sterols, sphingolipids and lyso-phospholipids are derived in part from acetyl CoA. A major transcriptional regulator of lipid metabolism-related genes is the sterol regulatory element binding protein (SREBP) family of transcription factors. Among the SREBP-targeted genes are ATP citrate lyase (ACLY), acetyl-CoA carboxylase 1 (ACC1), fatty acid synthase (FASN), stearoyl-CoA desaturase 1 (SCD) and fatty acid transporters. SREBP is processed and translocates to the nucleus to induce transcription of its target genes. mTOR modulates SREBP and other regulators and effectors of lipid metabolism (Figure 4). Rapamycin blocks the expression of genes involved in lipogenesis and prevents the nuclear accumulation of SREBP [110]. SREBPs are synthesized as inactive precursors that reside in the endoplasmic reticulum and translocate to the nucleus after processing from the Golgi. This active processed form induces transcription of SRE-containing genes. The processing step is thus sensitive to sterol levels and controlled by mTORC1 signaling. An enhancement of the processed forms of SREBP1 occur in the TSC-deficient cells [14]. S6K1 also regulates SREBP processing [111]. The sterol regulatory element was the most highly enriched DNA motif in a gene expression study of rapamycin-sensitive genes [14]. mTORC1 may also regulate SREBP via negative regulation of lipin1, a phosphatidic acid phosphatase that represses SREBP activity [112]. Lipin is phosphorylated by mTORC1 at multiple phosphosites including both rapamycin-sensitive and -insensitive sites [112]. When phosphorylated, lipin1 accumulates in the nucleus and represses SREBP-dependent gene transcription. mTORC1 signaling is necessary, but not sufficient to activate SREBP1 and lipid synthesis in the liver. mTORC2 is also required for SREBP1c activation and lipogenesis [113].

In B-cell non-Hodgkin lymphoma, both fatty acid synthesis and glycolysis are upregulated in a PI3K/mTOR-dependent manner. The fatty acid synthase FASN was overexpressed and thereby the lymphoma was more sensitive to the FASN inhibitor C75 than primary B cells. [114]. In hepatocellular carcinoma triggered by co-expression of Akt and c-Met, their growth was dependent on mTORC1 and fatty acid synthase (FASN) [115]. Carcinogenesis was prevented by FASN ablation. In neurofibromatosis type 2 (NF2) disorder, which is characterized by multiple tumors in the central nervous system such as schwannomas and meningiomas, the deficiency of the *Nf2* gene, which encodes the tumor suppressor Merlin, upregulated mTORC1 leading to elevated expression of key enzymes involved in lipogenesis. The inhibition or knockdown of FASN led to apoptosis of NF2-deficient cells [116]. FASN catalyzes the synthesis of palmitic acid from malonyl-CoA. When the Nf2-deficient cells were treated with compounds that blocked the production of malonyl CoA, the sensitivity to FASN inhibitors was reduced. In breast cancer, mTORC1 also promotes expression of stearoyl CoA desaturase 1 (SCD1), the rate limiting enzyme in monounsaturated fatty acid synthesis [117]. Rapamycin inhibited SCD1 promoter activity and decreased the expression of SREBP1 through a mechanism involving eIF4E.

In chronic myelogenous leukemia that are addicted to glucose metabolism for survival, inhibition of mTORC1 by rapamycin or S6K1 knockdown led to increased fatty acid oxidation and increased the expression of the fatty acid transporter carnitine palmitoyl transferase 1c (Cpt1c) [118]. This transporter is also linked to rapamycin resistance in human lung tumors [119]. The fatty acid binding protein 4 (FABP4), which is an adipokine for fatty acid transport increases breast cancer cell proliferation and promotes expression of fatty acid transporters such as CD36 and FABP5 [120]. Exogenous treatment of FABP4 increased Akt and MAPK signaling. The increased expression of the FA transporter CD36 in gastric cancer tissues also correlated with poor prognosis in patients [121]. CD36 mediates the palmitate acid-induced metastasis of GC via Akt. Together, these findings suggest that targeting fatty acid transport could have therapeutic benefits for cancers that rely on increased lipid metabolism for growth and proliferation.

In addition to the role of mTOR in fatty acid synthesis, it is also involved in lipid synthesis. Lipids are used not only for membrane biogenesis but also as signaling molecules. mTORC2 promotes the increased synthesis of sphingolipids and glycerosphospholipid, leading to liver steatosis and hepatocellular carcinoma [122].

### 2.5. Glutamine Metabolism

As with glucose metabolism, mTOR signaling impinges on multiple aspects of glutamine metabolism (Figure 5). On the other hand, mTOR is also sensitive to glutamine fluctuations and its activity is influenced by perturbations in glutamine metabolism. Glutamine is the most abundant non-essential amino acid in the plasma and is avidly used by proliferating tumor cells. It is acquired from the environment via transporters including Slc1A5 (ASCT2), Slc38A1, Slc38A2 or Slc38A5. The expression of these transporters is upregulated in many types of cancer [123]. Slc1A5 is a major transporter of glutamine in most cells. Knockdown of ASCT2 diminishes mTORC1 activity and tumor growth [124,125,126,127,128]. Although the absence of Slc1A5 did not significantly diminish intracellular Gln or Glu levels, it disrupted influx of leucine and diminished levels of other amino acids crucial for redox homeostasis [128]. Leucine, an essential amino acid, is acquired by the cell through counter-transport with Gln [21]. The expression of the heterodimeric glutamine antiporter Slc7A5/Slc3A2 (LAT/CD98) is associated with elevated mTORC1 activity in cancer [129].

Glutamine is a versatile molecule, as it serves as an alternative carbon source for energy production and its carbon and nitrogen are also used for biosynthetic reactions [130]. Hence, cancer cells can develop addiction to glutamine and upregulate its metabolism to provide necessary building blocks for the growing tumor [131,132]. mTOR has been linked to the regulation of many of these processes. Glutamine is a precursor for α-ketoglutarate (αKG) and is used to replenish TCA intermediates (anaplerosis) in proliferating cells. Glutamine is metabolized via glutaminolysis, which consists of two steps. The first step is catalyzed by glutaminase (GLS) and converts glutamine to glutamate. The second is catalyzed by glutamate dehydrogenase (GDH) and converts glutamate to αKG. Conversely, glutamine can be generated by glutamine synthetase (GS) which uses ATP and NH4^+^ to generate glutamine from glutamate. mTORC1 stimulates glutamine metabolism via regulation of transcription factors involved in expression of glutaminolysis-related genes. mTORC1 mediates translational upregulation of Myc via a mechanism involving the S6K1 eIF-4E [86,133]. Myc in turn represses transcription of miR-23a and miR-23b, which are both repressors of GLS [134]. Oncogenic signals such as elevated Myc levels increase glutamine uptake and metabolism through a transcriptional program that includes enhancement of expression of mitochondrial glutaminase [134,135]. Other signals independent of mTOR could also contribute to promoting glutaminolysis and could thus be exploited for co-targeting with mTOR inhibitors. In lung squamous cancer cell carcinoma that underwent chronic mTOR inhibition and suppression of glycolysis, glutaminolysis was enhanced via a mechanism involving GSK3. Increased glutaminolysis and/or increased levels of GLS have been found in a number of cancers that become resistant to other therapies. Combined treatment with the glutaminase inhibitor CB-839 and mTOR inhibitors shows efficacy in overcoming therapeutic resistance to other targeted inhibitors in these different types of cancer [136,137]. Another transcription factor that is regulated by mTORC1 to promote glutaminolysis is CREB2 (cAMP-responsive element binding 2). mTORC1 promotes the proteasome-mediated degradation of CREB2 and represses SIRT4 transcription [138]. SIRT4 negatively regulates GDH by ADP-ribosylation. Thus, mTORC1 promotes glutamine metabolism via negative regulation of CREB2, which ultimately leads to activation of GDH.

mTORC2 is also emerging to play a crucial role in glutamine metabolism. The expression of Myc in glioblastoma is mTORC2-dependent [107]. Furthermore, knockdown of rictor decreases levels of αKG that are likely derived from glutaminolysis, suggesting that mTORC2 could regulate this process as well [97]. However, inhibition of PI3K or Akt in glioma that expressed oncogenic levels of Myc did not affect glutaminolysis in these tumors [135]. A possibility is that the overexpression or deregulation of Myc uncouples it from PI3K/Akt signals.

While mTOR modulates glutamine metabolism, the latter also reciprocally modulates mTOR. Glutamine, in combination with leucine activates mTORC1 by enhancing glutaminolysis and αKG production. Glutaminolysis correlates with increased mTORC1 activity and is necessary for GTP loading of RagB and activation of mTORC1 signaling. It also promotes cell growth and inhibits autophagy via regulation of mTORC1 [29]. The import of glutamine by the transporter Slc1A5 has also been suggested to be the rate-limiting step that activates mTOR. On the other hand, glutamine depletion that can occur as an off target effect of using asparaginase, which has glutaminase activity, indirectly inhibits mTOR activity via decreased leucine uptake in AML [126]. Glutamine synthase levels also affect mTORC1 activation. In liver cancer with mutations in β-catenin, levels of GS, a target of β-catenin, are increased and its disruption prevents mTOR phosphorylation at Ser2448, suggesting downregulation of mTOR activity [139]. In some cell types, in the absence of glutamine, cells utilize ammonia as an alternative nitrogen source. This ability to utilize ammonia is linked to mTORC1 as well as GDH and AMPK [140]. mTORC2 also responds to levels of glutamine or its catabolites. It is interesting to note that glutamine depletion decreases mTORC1 activity while increasing mTORC2 activity [54,141]. This differential regulation of mTORC1 and mTORC2 is mediated by sestrin2 and allows survival of these cancer cells by maintenance of energy and redox balance [141]. However, inhibition of glutaminase or GDH did not affect Akt phosphorylation while it diminished mTORC1 activation [29]. How glutaminolysis affects mTORC2 requires further investigation.

Glutamine is also used for the production of UDP-GlcNAc, a metabolite produced by the hexosamine biosynthesis pathway (HBP), which, in turn is used for protein and lipid glycosylation [142]. During glucose or glutamine starvation, mTORC2 promotes the expression and phosphorylation of glutamine fructose-6-phosphate amidotransferase-1 (GFAT1), the rate limiting enzyme involved in the de novo hexosamine biosynthesis, in order to maintain flux through the HBP [54,143]. The expression of GFAT1 is dependent on the levels of glutamine catabolites such as α-KG. In turn, mTORC2 also promotes the generation of glutamine catabolites. Maintaining flux through the HBP via glucosamine supplementation, which bypasses the GFAT1-catalyzed reaction, during glucose starvation could rescue Akt signaling in the absence of insulin [144]. These findings revealed a feedback relationship between mTORC2 and the HBP.

### 2.6. Pentose Phosphate Pathway and Nucleotide Synthesis

The pentose phosphate pathway (PPP) generates reducing equivalents in the form of NADPH and ribose-5-phosphate for nucleic acid synthesis. The rate-limiting reaction in the PPP is catalyzed by Glucose-6-phosphate dehydrogenase (G6PD) [145]. PPP consists of two stages: an irreversible oxidative and a reversible nonoxidative stage (Figure 6). In the oxidative reactions, the G6PD-catalyzed reaction produces NADPH. The non-oxidative stage is a series of reversible reactions, converting glycolytic intermediates into ribose-5-phosphate, a key precursor for DNA and RNA synthesis. The PPP ultimately generate phosphoribosyl pyrophosphate (PRPP), the precursor for nucleotide synthesis. Cancer cells have increased G6PD levels and PPP activity [146,147,148,149]. Whereas pentose phosphates are essential for nucleotide production, NADPH serves as a reducing agent in several synthetic steps of fatty acid, cholesterol, and steroid hormones, along with detoxification reactions. Hence, blocking G6PD could have potent anti-tumor effects by preventing nucleotide synthesis as well as impairing redox balance. The G6PD competitive inhibitor, 6-aminonicotinamide (6-AN) is being used to target cancer cells with increased PPP activity and has shown antineoplastic effects [150,151]. Downregulating G6PD expression has been shown to decrease cell viability of bladder cancer cell lines due to accumulation of ROS [149]. mTORC1 can stimulate flux through the oxidative branch [14]. mTORC1 is involved in this pathway via transcription of genes encoding enzymes that drive the PPP. By regulating expression of these genes, mTORC1 promotes production of ribose-5-phosphates, which are used in purine and pyrimidine nucleotide synthesis and production of NADPH.

Cancer cells require an abundant pool of purines and pyrimidines, which serves as building blocks for DNA and RNA synthesis. Nucleotide synthesis utilizes two pathways: the de novo synthesis pathway, which assembles complex nucleotides from basic molecules and the salvage pathway, which generates nucleotides from degradation intermediates. mTORC1 is involved in the production of pyrimidines via de novo pathways [152,153]. mTORC1 promotes activation of CAD (carbamoyl-phosphate synthetase 2, aspartate transcarbamylase, and dihydro-orotase) via S6K1. S6K1 phosphorylates CAD on Ser1859 [153]. CAD catalyzes the initial steps of pyrimidine synthesis by utilizing glutamine, bicarbonate, and aspartate to generate pyrimidine rings. De novo pyrimidine synthesis is enhanced by mTORC1 and S6K in response to growth factor or amino acid stimulation, although they are not essential for de novo synthesis per se [152]. mTORC1 also promotes purine nucleotide biosynthesis. Purines are assembled on the ribose sugar PRPP, wherein carbon and nitrogen molecules are donated by non-essential amino acids and one-carbon formyl units from the tetrahydrofolate (THF) cycle. mTORC1 has been implicated in the increased expression of phosphoribosyl pyrophosphate synthase 2 (PRPS2), the rate-limiting enzyme that provides PRPP for both purine and pyrimidine nucleotides, in Myc-transformed cells [154]. mTORC1 also promotes transcription of numerous enzymes contributing to purine synthesis including the mitochondrial tetrahydrofolate cycle enzyme methylenetetrahydrofolate dehydrogenase 2 (MTHFD2). MTHFD2 expression is induced by mTORC1 via ATF4 [155].

mTORC2 is also involved in modulating the PPP. In an insulin-driven model of hepatocellular carcinoma cells, Akt drove the upregulation of the PPP through several mechanisms, including via increase of phosphate dehydrogenase and ribose 5-phosphate isomerase A expression and activity, as well as through driving glycolysis [156]. The role of mTORC2 in the PPP is further supported from studies in yeast wherein the yeast TORC2 physically interacted with proteins that play a role in the PPP [157]. Metabolic intermediates such as 6-phospho-D-gluconate (6PG) and ribose-5-phosphate are strongly downregulated in response to TOR2 inhibition, suggesting that TORC2 could post-translationally regulate PPP modulators.

mTORC2 also regulates purine synthesis via Akt [158]. The PI3K/Akt signaling axis regulates the early steps of the non-oxidative PPP at the level of PRPP synthesis and later steps by modulating the activity of aminoimidazole-carboxamide ribonucleotide transformylase IMP cyclohydrolase (ATIC) [158]. Akt also phosphorylates transketolase (TKT) on Thr382, a key enzyme of the nonoxidative PPP, leading to increased flux through this pathway and increasing purine synthesis [159].

### 2.7. Other Metabolic Pathways

mTORC1 also promotes serine biosynthesis and one-carbon pathway metabolism. The product of this pathway, S-adenosylmethionine (SAM), which is derived from methionine, is also sensed by mTORC1 via SAMTOR [160]. In mouse models and primary pancreatic epithelial cells, the oncogenic cooperation of KRAS activation and LKB1 loss led to mTOR-induced activation of the serine-glycine-one carbon pathway that enhances generation of S-adenosyl methionine (SAM) [161]. This was accompanied by increased expression of DNA methyltransferases that consequently modified the epigenome to promote tumorigenesis. In de novo or during therapy-induced neuroendocrine prostate cancer (NEPC), downregulation of PKCλ/ι upregulates serine biosynthesis through mTORC1 and ATF4-dependent pathway [162]. This metabolic reprogramming generates increased SAM that is utilized for epigenetic changes that promote NEPC attributes.

## 3. Targeting the mTOR Pathway in Cancer

Due to the critical role that mTOR plays in cell growth and metabolism and availability of the natural compound, rapamycin, to inhibit its activity, there are numerous ongoing efforts to target the mTOR signaling pathway for cancer therapy. Various studies have already shown that deregulation of the mTOR pathway is present in many cancers. For example, mutations causing an increase in mTOR activity have been found in solid tumors [163,164,165]. Additionally, mutations in upstream regulators of mTOR including the oncogene *PIK3CA*, which encodes PI3K and the tumor suppressor *PTEN* occur frequently in human tumors [166]. Moreover, genetic lesions can also promote the activation of the PI3K/mTOR pathway in cancer cells, such as those encoding for Ras, Akt, TSC1/2, Notch1, and receptor tyrosine kinases [167]. Mutations and genomic alterations in metabolic enzymes and other key regulators of metabolic pathways of which mTOR has been linked are also common in cancers [71,168]. The upregulation of the mTOR pathway due to one or more of these mutations make mTOR an attractive target in tumors. However, due to other signaling molecules that also play a role in the control of metabolism as well as the presence of de novo and salvage synthesis of metabolites, there is a need to design appropriate combination therapeutic strategies for more effective cancer treatment. Here, we discuss the current strategies to inhibit the mTOR pathway and combination therapy that could more effectively block critical metabolic pathways and other signaling molecules that control metabolism. The remarkable success of immunotherapy, which utilizes a patient’s own immune system to combat tumors, also underscores the importance of considering not just the tumor’s metabolism but also of neighboring cells, including immune cells. We include a discussion of how modulating mTOR signaling has implications in immunotherapeutic approaches.

### 3.1. Rapalogs: Targeting mTORC1 Activity

Rapamycin and its analogs (rapalogs) are the first generation of mTOR inhibitors, which selectively inhibit the activity of mTORC1 by binding to FKBP-12 and forming a ternary complex with mTOR. Rapamycin is an allosteric inhibitor of mTOR and it inhibits some of the functions of mTORC1, such as phosphorylation of the protein kinase S6K1. As discussed above, S6K1 has numerous downstream targets involved in protein synthesis as well as other metabolic targets [59]. The clinical use of rapamycin is limited due to poor water solubility and stability. Thus, several pharmaceutical companies have developed rapamycin analogs with improved pharmacokinetic properties (Table 1). Rapalogs differ in their chemical properties in terms of drug solubility and metabolism. For example, temsirolimus (Torisel, CCI779; Wyeth) and ridaforolimus (AP23573, deforolimus, Merck/ARIAD) are water soluble and must be administered intravenously [169,170]. Rapalogs have been undergoing clinical trials for various malignancies and have already been approved by the Food and Drug Administration (FDA) for the treatment of specific types of cancers. Everolimus (Affinitor, RAD001; Novartis) has been efficacious and FDA-approved for the treatment of advanced renal cell carcinoma that progressed after treatment with sunitinib and/or sorafenib [171,172]. Everolimus has also been approved for treatment of both subependymal giant-cell astrocytoma (SEGA) and renal angiomyolipoma with tuberous sclerosis complex [173,174,175]. Additionally, everolimus significantly prolonged progression-free survival (PFS) among patients with advanced pancreatic neuroendocrine tumors and was associated with a low rate of severe adverse events in a Phase III study, making everolimus the first new treatment for this type of cancer in almost 30 years [176]. Furthermore, everolimus is the first medication approved to treat progressive, nonfunctional lung and gastrointestinal tumors [177]. Temsirolimus (Torisel, CCI779; Wyeth) is another rapalog and is currently approved for advanced renal cell carcinoma [178,179] and was shown to significantly improve PFS in patients with relapsed or refractory mantle cell lymphoma [180].

There are numerous clinical trials on rapalogs as monotherapy that have yielded some promising results for the treatment of other types of cancer. Results from a Phase II study of everolimus in patients with advanced follicular -derived thyroid cancer yielded favorable results, resulting in clinically relevant antitumor activity and a relatively low toxicity profile [181]. Ridaforolimus is the newest rapalog and has been investigated in treating sarcomas. Ridaforolimus had an overall PFS rate at 6 months of 23.4% for soft tissue and bone sarcomas, including primary bone sarcomas, leiomyosarcomas, and liposarcomas in a phase II study [182]. However, an international phase III trial of ridaforolimus showed that it delayed tumor progression only to a small degree in patients with metastatic sarcoma [183]. Ridaforolimus displayed anti-tumor activity in advanced endometrial cancer patients in phase II clinical trials though with associated significant toxicity [184,185]. A phase I study of ridaforolimus in pediatric patients with advanced solid tumors had favorable toxicity and shows promise for combination therapies [186]. A phase II study of ridaforolimus in women with recurrent or metastatic endometrial cancer showed tolerability and modest activity of the drug [187].

In many types of cancer however, rapalogs do not provide marked benefits or only stabilize the disease. For example, everolimus did not significantly improve overall survival in comparison to supportive care in previously treated gastric cancer [190]. Another study that involved patients with advanced hepatocellular carcinoma showed that everolimus did not improve overall survival when their disease progressed during or after receiving sorafenib (inhibitor of growth factor receptors and Raf) or who were intolerant of sorafenib [191]. Everolimus also failed to produce clinically relevant patient response in a phase II trial for patients with relapsed or cisplatin refractory germ cell tumors [192]. In a phase II trial of everolimus in patients with previously treated recurrent or metastatic head and neck squamous cell carcinoma, there was no efficacy likely due to toxicity [193]. In a phase II study of temsirolimus in women with platinum-refractory/resistant ovarian cancer or advanced recurrent endometrial carcinoma, the temsirolimus treatment was tolerated and a few patients had long-lasting stable disease [195].

The efficacy of rapalog treatment as a monotherapy has been modest is not surprising given results from pre-clinical studies that rapamycin only inhibits a subset of mTORC1 substrates [234]. Whereas rapamycin inhibits S6K1, it does not fully inhibit 4EBP1 phosphorylation, thus ineffective in blocking cap-dependent translation in most cell types [235]. Phosphorylated 4EBP1 inhibits the pro-oncogenic eIF4E. eIF4E-mediated translation are upregulated in tumors and blocking this pathway may be crucial to preventing tumor growth in specific cancers [236,237,238]. As discussed below, mTOR inhibitors that could block the catalytic activity of mTOR could more effectively inhibit mTOR functions and may have better anti-tumor activity.

Despite the limited efficacy of rapalogs as a treatment for a variety of cancers, they remain promising for particular types of cancers. In a few studies, tumors that have upregulated mTORC1 due to inactivation of TSC displayed sensitivity to rapalog treatment. Using whole genome sequencing to analyze molecular determinants of sensitivity to everolimus treatment in metastatic bladder cancer, mutations in TSC1 were uncovered [239]. In another study, treatment with oral sirolimus led to significant reductions in cardiac rhabdomyomas in infants [188]. These rhabdomyomas are associated with tuberous sclerosis complex. In a phase II trial of everolimus on patients with thyroid cancer, whole-exome sequencing of tumors prior to treatment revealed that in a responding patient, there was a mutation in TSC2 that is known to inactivate it as well as a mutation in another tumor suppressor, folliculin 1 (FLCN) that also results in increased mTORC1 signaling [194]. Another study of outlier cases in responders who had metastatic renal cell carcinoma treated with rapalogs also revealed that the strong responders had alterations in TSC1 and mTOR, leading to increased mTORC1 signaling [240]. Importantly, as mentioned above, everolimus has been approved for the treatment of SEGA and renal angiomyolipoma with tuberous sclerosis complex. Altogether, these findings suggest that tumors with increased mTORC1 activation due to TSC mutations may be vulnerable to rapalog treatment. Hence, pre-screening patients for TSC mutations may provide information on the possible benefits of rapalog therapy. However, advanced or highly metastatic tumors, despite having an upregulated mTORC1 signaling, may not necessarily be responsive to rapalog monotherapy due to additional mutations that deregulate other compensatory pathways, as further discussed below [241].

### 3.2. Co-targeting mTORC1 and Growth Factor Signaling

A potential hurdle in the use of rapalogs for cancer treatment is the existence of feedback mechanisms that could occur when mTORC1 signaling is downregulated. Growth factors potentiate mTORC1 signals via mechanisms that are dependent and independent of TSC. However, to prevent excessive signaling to mTORC1 that could lead to deregulated growth and proliferation, growth factor signals are subjected to negative feedback regulation by mTORC1 [85]. Activated S6K1, an mTORC1 target, reduces signals from the insulin receptor (IR) by inhibitory phosphorylation of insulin receptor substrate-1 (IRS-1), thus mitigating signals downstream of the insulin receptor (IR). Rapamycin treatment, which inactivates S6K1, can thus prevent the suppression of insulin-PI3K signaling. Since this effect has also been observed in other tyrosine kinase receptors (RTK), rapamycin could thus have the undesirable effect of upregulating growth factor/PI3K signaling. In addition, rapamycin treatment has also been shown to increase expression of growth factor receptors such as IGF1R [242]. Furthermore, RTKs not only signal to PI3K/mTOR but also transduce signals to other signaling pathways such as the Ras/MAPK pathway, that play a role in growth and proliferation [243]. Therefore, combining rapamycin treatment with inhibition of RTKs could serve as a more effective strategy for inhibiting mTOR and tumor growth.

Rapalogs have been used in combination with inhibitors of receptor tyrosine kinases (Table 2). Several clinical trials using rapalogs as combination therapy have been conducted in breast cancer, wherein HER2, a member of the epidermal growth factor receptor (EGFR), is often expressed or amplified. BOLERO-3 trial found that the addition of everolimus to trastuzumab, an anti-HER2 antibody, and vinorelbine (mitotic inhibitor) prolonged PFS in patients with trastuzumab-resistance, taxane-pretreated, and HER2-positive breast cancer [244]. A phase II study showed that ridaforolimus with trastuzumab demonstrated anti-tumor activity for patients with HER2+ trastuzumab-refractory breast cancer [245]. In a phase I study, the combination of neratinib, a small-molecule irreversible pan-HER tyrosine kinase inhibitor, and temsirolimus displayed anti-tumor activity in patients with HER2+ breast cancer resistant to trastuzumab, HER2-mutant non-small cell lung cancer, and tumors without identified mutations in the HER-PI3K-mTOR pathway [246]. In a phase I study of temsirolimus with the c-Met inhibitor tivantinib for the treatment of various advanced solid tumors, the combination treatment was well tolerated and had some clinical activity [247]. In a phase I study of combination temsirolimus with cetuximab (anti-EGFR monoclonal antibody) for treatment of advanced solid tumors, clinical activity was modest and thus not further pursued [248]. A combination of temsirolimus with the VEGF inhibitor bevacizumab and cetuximab had partial responses in head and neck squamous cell carcinoma with numerous toxicities [249]. In a phase II study of temsirolimus with bevacizumab for patients with metastatic renal cell carcinoma previously treated with other VEGFR TKI, the combination treatment resulted in modest activity and dose reductions were needed due to toxicity [250]. Another phase II study of lower dose ridaforolimus with dalotuzumab, which inhibits autophosphorylation of IGF1R in ER+ advanced breast cancer had similar efficacy but with higher incidence of adverse events compared to treatment with the aromatase inhibitor, exemestane [251].

However, the combination of cixutumumab, an anti-IGF-1R antibody, with temsirolimus did not show objective responses in pediatric and young adults with refractory or recurrent sarcoma [252]. A randomized phase II trial of ridaforolimus with the IGF1R inhibitor dalotuzumab and exemestane (R/D/E) was compared with R/E in patients with advanced breast cancer. The PFS of the R/D/E did not improve compared to R/E, likely due to lower doses of ridaforolimus in the R/D/E arm. While there was less toxicity, the efficacy of the R/D/E regimen was poorer [280]. Taken together, rapalogs in combination with growth factor receptor inhibitors have limited efficacy.

Despite the limited efficacy and increased toxicity of combined rapalog/RTK inhibitors, exceptional cases of strong responders as well as pre-clinical studies may provide insight on how this strategy can yield more positive results. In one study, tumors that may have activating mutations of mTOR may potentially be more sensitive to rapalog treatment. A mutation in the kinase domain of mTOR (E2419K) and another at the FRB domain (E2014K) activated mTORC1 signaling in a patient who had a strong response during a phase I trial of everolimus in combination with pazopanib, an inhibitor of growth factor receptors [164]. Since mTOR is also part of mTORC2, such mutation in the kinase domain may also affect the growth factor-modulated mTORC2 signaling. Hence, the molecular alterations that may be associated with vulnerability to combined rapalog/RTK inhibition warrant further investigation. In mouse models of uterine serous carcinoma, combined ridaforolimus and HER2 blockade with lapatinib/trastuzumab had a better anti-tumor activity when tumors had PIK3CA (E542K) mutations together with HER2 gene amplification rather than those without PIK3CA mutations. While lapatinib downregulated mTORC2 signaling in the former tumors, only the combined ridaforolimus/lapatinib treatment was able to abrogate S6 phosphorylation in such tumors, correlating with the anti-tumor activity [253]. These findings reveal how both mTORC1 and mTORC2 signaling downregulation is critical to achieve better efficacy when both mTOR complexes are upregulated either due to amplified RTK or mutated PIK3CA.

### 3.3. Targeting mTORC1 and mTORC2 with ATP-Competitive mTOR Kinase Inhibitors (TORKIs)

The second generation of mTOR inhibitors differs from rapamycin and rapalogs by the fact that they directly bind to the ATP-binding site of the kinase domain of mTOR. Furthermore, unlike rapamycin and its analogs that allosterically inhibit mTOR and block mTORC1 function, ATP inhibitors can directly inhibit the enzymatic activity of both mTORC1 and mTORC2. They are also more specific to mTOR and have an IC_50_ that is much lower than that for PI3K. However, it is worth noting that the well characterized substrates of mTOR, such as S6K1 (for mTORC1) and Akt (for mTORC2) are only allosterically regulated by mTOR via phosphorylation at their hydrophobic and/or turn motif sites [281]. The activation of these AGC protein kinases via phosphorylation at their catalytic loop occurs via PDK1. Hence, blocking mTOR catalytic activity would only partially inactivate these AGC kinases and may not fully block their functions. Nevertheless, this class of mTOR inhibitors displayed anti-proliferative and cytotoxic effects in preclinical studies [282]. Inhibition of mRNA translation and induction of cell cycle arrest, apoptosis, and autophagy was seen in the treatment of acute myeloid leukemia and T-cell acute lymphoblastic leukemia cells with ATP mTOR inhibitors [283,284]. In another study, hepatocellular carcinoma cells treated with AZD8055 (AstraZeneca) led to cell death, induction of autophagy, and activation of AMPK [196]. AZD8055 was also studied on Hep-2, a human laryngeal cancer cell line. In these cells, expression of mTOR was downregulated after treatment with AZD8055. Furthermore, pro-apoptotic factors, including Bax and Caspase3, were up-regulated with AZD8055 treatment and anti-apoptotic factor Bcl-2 was reduced. This suggests that AZD8055 can inhibit proliferation and induce apoptosis in Hep-2 cells [197].

Although pre-clinical evidence was encouraging, early clinical trials have had mixed results. Phase I trials with AZD8055 led to elevated transaminases in patients with advanced solid malignancies and lymphoma, displaying an unfavorable toxic profile. Therefore, AZD8055 is not being developed further, and a follow-up compound, AZD2014 (Vistusertib, AstraZeneca) that has no rise in transaminases is being investigated instead [198,199]. A phase I trial with AZD2014 showed a more favorable pharmacological profile overall and demonstrated efficacy as a single agent in heavily pre-treated solid tumors [200]. However, a phase II trial that studied AZD2014 versus everolimus found AZD2014 to be inferior in treating patients with VEGF-refractory renal cell carcinoma and led to early termination of the study [201]. Kinome profiling of samples from patients with advanced-stage ovarian clear cell carcinoma (OCCC) revealed increased alterations in the PI3K/Akt/mTOR pathway in about 91% of tumors [285]. The majority of the OCCC cell lines tested displayed more sensitivity to mTORC1/2 inhibition (AZD8055) than to drugs targeting the ERBB family of RTKs or to inhibitors of DNA repair signaling.

Sapanisertib (TAK-228, INK-128, Millennium Pharmaceuticals) is another ATP-competitive mTOR kinase inhibitors and has shown some promising results in phase I trials. Sapanisertib and paclitaxel with or without trastuzumab was evaluated in patients with advanced solid malignancies. The most common types of cancers in this study were lung (21%), ovarian (12%), and breast, endometrial, and esophageal (9% each). Sapanisertib was well tolerated in this study and exhibited anti-tumor activity in a range of tumor types. Out of 54 patients, eight experienced partial responses and six had stable disease lasting over 6 months [204]. Sapanisertib was also well tolerated and displayed preliminary therapeutic activity in patients with refractory multiple myeloma, non-Hodgkin’s lymphoma, and Waldenström’s macroglobulinemia. Almost half of the patients in the study achieved stable disease [205]. OSI-027 (ASP7486, Astellas Pharma) is another ATP-competitive mTOR kinase inhibitor currently undergoing clinical trials. A recent phase I trial of OSI-027 showed inhibition of mTORC1/mTORC2 in patients with advanced tumors. However, disturbances of renal function were common, and doses above tolerable levels in two of the tested drug schedules were required for sustained biological effects [206]. Another TORKI, MLN0128, was used in a phase II clinical trial on metastatic castration-resistant prostate cancer but had limited clinical efficacy due to dose reductions as a consequence of toxicity. There was poor inhibition of mTOR signaling targets such as Akt and 4EBP1 and compensatory increase of androgen receptor activity [207].

These early clinical trials have yielded mixed results with the negative outcome due to dose limiting toxicities. It will be important to identify predictive biomarkers to determine which patients could benefit from the TORKIs. For example, in gastric cancer and small cell lung cancer tumors that had amplification of rictor expression, there was specific sensitivity to AZD2014 [202,203]. The toxicity of the TORKIs also highlights the important role of mTOR in metabolic homeostasis of normal proliferating cells. Hence, revisions of formulations or treatment regimen may also be refined to avoid toxicities while obtaining a more durable response. More highly selective mTOR inhibitors are also currently being developed to improve specificity and metabolic stability [286].

### 3.4. Dual PI3K/mTOR Inhibitors: Targeting PI3K and mTOR Signaling

The PI3K signaling cascade is deregulated in many types of human cancer [287]. Upon growth factor stimulation, PI3K is activated and phosphorylates the phosphatidylinositol-4,5 bisphosphate (PIP2) to generate phosphatidylinositol-3,4,5 trisphosphate (PIP3). Increased PIP3 levels recruit a number of signaling molecules to the membrane periphery, including Akt. Signals from PI3K are antagonized by the tumor suppressor PTEN, which contains a lipid phosphatase domain that catalyzes conversion of PIP3 to PIP2. PTEN is often mutated or inactivated in several cancers. Other common mutations are those affecting the gene coding for p110α, a catalytic domain of PI3K. The p110 subunit of PI3K and catalytic domain of mTOR are structurally similar. Therefore, dual inhibitors of PI3K and mTOR that target their catalytic domains have been engineered [6]. These dual inhibitors would block not only PI3K activity but both mTORC1 and mTORC2 complexes and would thus have broader inhibitory capabilities than rapamycin. Furthermore, Akt would also be suppressed by these dual inhibitors since blocking PI3K would diminish production of PIP3, the lipid that acts as a docking site for Akt and PDK1. Inhibition of mTORC2 would also block the allosteric activation of Akt. Hence, dual targeting of PI3K and mTOR would have more extensive inhibitory effects on the PI3K/mTOR pathway.

Pre-clinical studies using cell and mouse models demonstrated the efficacy of dual PI3K/mTOR inhibitors such as BEZ235 or dactolisib (NVP-BEZ235, Novartis) in several types of cancers including breast cancer, lung adenocarcinomas, glioblastoma, [209,210,211]. In these models, tumors with mutations in PIK3CA (catalytic subunit) or increased PI3K/mTOR pathway activation displayed specific sensitivity to the inhibitor. In another study of HER2-amplified with or without PIK3CA mutation breast cancer cells, dactolisib induced cell death and apoptosis by activating caspase-2 and PARP cleavage [212]. In a multiple myeloma model, dactolisib prevented phosphorylation of mTORC1/2 targets and induced cell cycle arrest [213]. Gedatolisib (PKI-587; Pfizer) had better efficacy than everolimus in inhibiting proliferation of gastroenteropancreatic neuroendocrine (GEP-NENs) tumor cell lines [221]. The inhibitor promoted cell cycle arrest and induced apoptosis and prevented phosphorylation of 4EBP1. Hence, in pre-clinical studies, dual PI3K/mTOR inhibitors are more effective at promoting tumor cell death, as expected.

However, early clinical trials of dactolisib have not shown as much efficacy. In an early Phase I study, BEZ235 given to patients with advanced solid tumors led to dose limiting toxicity [214]. In one study, patients with advanced renal cell carcinoma received escalating doses of dactolisib had high incidence of significant toxicities across all dose levels tested with no objective response. Based on these results, dactolisib is not recommended for patients with renal cell carcinoma [215]. Dactolisib was also associated with a poorer tolerability profile and was not more effective than everolimus in mTOR inhibitor-naïve patients with pancreatic neuroendocrine tumors [216]. In another Phase I/IB study of BEZ235 in patients with advanced solid tumors including those with advanced breast cancer, the inhibition of PI3K/mTOR was not sufficient and adverse effects were prevalent [217]. In a phase II study of BEZ235 in patients with locally advanced or metastatic transitional cell carcinoma, clinical activity was modest and was accompanied by considerable toxicities [218]. Combination therapy utilizing dactolisib has also been disappointing. Treatment strategies utilizing dactolisib with everolimus demonstrated limited efficacy and tolerance in patients with advanced solid tumors [219]. In patients with castration-resistant prostate cancer, treatment with dactolisib and abiraterone acetate, an anti-androgen agent, also yielded poor efficacy and tolerance profile [220].

Other dual PI3K/mTOR inhibitors are also undergoing clinical trials. Apitolisib (GDC-0980, Genentech) is another dual inhibitor of PI3K/mTOR. A phase I study in patients with advanced solid tumors revealed tolerability at 30 mg, with modest but durable anti-tumor activity [223]. Apitolisib was compared against everolimus in patients with clear-cell metastatic renal cell carcinoma in a phase II trial. Not only were adverse events more frequent in patients receiving apitolisib compared with everolimus, but apitolisib was also found to be less effective [224]. In a phase II study of apitolisib for the treatment of recurrent or persistent endometrial carcinoma (MAGGIE study), the anti-tumor activity was limited by the dose toxicity particularly for patients with diabetes. However, molecular profiling of evaluable archival tumor samples revealed that the patients with confirmed response had at least one alteration in a gene involved in the PI3K pathway [225]. In a phase II trial of voxtalisib (SAR245409, XL-765, Sanofi), there was acceptable safety profile with promising efficacy in patients with follicular lymphoma but limited efficacy in patients with mantle cell lymphoma, diffuse large B-cell lymphoma, or chronic lymphocytic leukemia [226]. The limited efficacy in the aggressive lymphomas could be due to incomplete blockade of PI3Kδ. While there seemed to be no strong correlation between the presence of PI3K or PTEN mutations and the response to voxtalisib, there was one patient with a PIK3CA and KRAS mutation who had a complete response.

There have been some promising trials with dual PI3K/mTOR inhibitors. In a phase II study in patients with recurrent endometrial cancer, gedatolisib demonstrated moderate activity [222]. Another study using BGT226 (NVP-BGT226, Novartis) was generally well tolerated with only three patients out of 57 having dose-limiting toxic effects. The study also reported 30% of patients reaching stable disease with nine achieving stable disease for over 16 weeks and 53% of patients reached stable metabolic disease. However, they found inhibition of the PI3K pathway to be inconsistent, but this may have been due to low systemic exposure [227].

It is interesting to note that the dual inhibition of PI3K and mTOR in patients displayed more toxicity than combined rapalog/growth factor receptor blockade. The latter strategy should, in theory, block PI3K signaling, at least in cases wherein PI3K is not constitutively active or PTEN is inactivated. Yet, it is likely less toxic since the amplified expression of RTKs occurs in specific tissues (e.g., HER2 in breast). In contrast, increased mTOR and PI3K signaling would be pervasive not only in tumors but highly proliferating normal cells. Hence, more studies are needed to determine if dual PI3K/mTOR inhibitors would be most effective for tumors displaying hyperactive PI3K/mTOR signaling such as those harboring PTEN inactivating or PI3K- and/or mTOR-activating mutations. These types of tumors may be more sensitive to lower doses of the dual inhibitor and would thus have increased efficacy with lower toxicity.

### 3.5. Targeting mTORC2 Signaling

Over-activation of Akt has been found to be associated with many types of cancer, including HER2-amplified breast cancer and glioblastoma [288,289]. mTORC2 modulates Akt signaling by allosteric phosphorylation of Akt on Ser473. mTOR as part of mTORC2 requires the presence of its partners rictor and SIN1 to function. Rictor has been found to be highly expressed in certain types of cancers, such as colorectal cancer and non-small cell lung cancer [208,290]. Therefore, there has been interest in using rictor as a target for cancer therapeutics.

In HER2-enriched breast tumors, rictor expression was upregulated significantly compared to non-malignant tissues. Genetic ablation of *rictor* led to decreased cell survival and phosphorylation at S473 on Akt as well as decreased tumor formation and tumor multiplicity in a HER2/Neu mouse model of breast cancer. In the same study, it was found that there was decreased Akt activation and cell survival in multiple HER2-amplified human breast cancer cell lines with rictor loss [288]. mTOR inhibition with rapamycin was demonstrated to induce activation of Akt in human gastric and pancreatic cancer cells. Knockdown of rictor upon rapamycin treatment in these cancer cells led to diminished Akt phosphorylation and function, impaired cell motility and potentiated the anti-migratory properties of rapamycin. This suggests that simultaneous inhibition of mTORC1 and rictor could be an approach to reduce metastatic spread of tumors [291]. Another study investigated the effects of RNAi-mediated gene silencing of both rictor and EGFR in glioblastoma cells. Here, it was also demonstrated that a combined approach can reduce cell migration. Furthermore, siRNA-mediated silencing of EGFR and rictor also increased sensitivity to irinotecan, temozolomide, and vincristine in PTEN mutant human GBM cell line U251MG and increased sensitivity to vincristine and temozolomide in LN229 cell line. Silencing of both EGFR and rictor also caused complete eradication of tumors in U251MG cell line [289]. A nanoparticle-based RNAi therapeutic that was engineered to target rictor was shown to decrease breast cancer cell growth and survival via intratumoral and intravenous delivery. Furthermore, using this molecule in combination with lapatinib led to an even greater reduction of tumor growth [292]. JR-AB2-011 is a small molecule inhibitor that has been recently developed, which prevents the interaction of rictor with mTORC2. This molecule demonstrated significant anti-tumor effects in glioblastoma xenograft studies [230].

Although there have not been human clinical trials yet to test the effectiveness of rictor inhibition, patients with rictor amplification may benefit from ATP-competitive mTOR kinase inhibitors that block both mTORC1 and mTORC2. One study found that a patient with rictor-amplified non-small cell lung cancer achieved tumor stabilization for 12 months with CC-223 (Celgene). Disease rapidly progressed when treatment with CC-223 was ceased [208]. In small-cell lung cancer cell lines with rictor amplification, there was increased sensitivity to ATP-competitive mTOR kinase inhibitors [203]. Furthermore, a gastric-cancer patient derived cell line with rictor amplification was found to be most sensitive to AZD2014 [202]. The expression of SIN is also upregulated in medullary and aggressive papillary thyroid carcinomas [293]. The increased SIN1 expression was associated with enhanced Akt activation. Future studies should reveal whether increased levels of mTORC2 components could serve as predictive biomarkers for sensitivity to TORKIs or more specific mTORC2 inhibitors.

### 3.6. RapaLink1

Another class of mTOR inhibitor, RapaLink, has been recently developed [231]. RapaLink consists of a rapamycin-FRB binding element linked to mTOR kinase inhibitor (TORKI). This inhibitor was generated to combine the advantages while overcoming the limitations posed by each of the two types of inhibitors. While rapamycin has limited inhibitory capacity, it is more stable in cells due to its binding to FKBP12. On the other hand, the TORKI have better blockade of mTOR but has poor durability. Thus, this new bivalent inhibitor combines the durable effect of rapamycin and the superior inhibitory capacity of TORKI. Furthermore, it crosses the blood–brain barrier and was able to block in vivo glioblastoma models [232]. Follicular lymphoma, an incurable form of B cell lymphoma, with a genetic mutant of the epigenetic regulator, EZH2, was sensitive to Rapalink1 [233]. The increased mTORC1 activity due to EZH2 mutant repression of Sestrin1, which acts as a tumor suppressor, likely contributed to the sensitivity of these tumors to mTOR inhibition. Future studies should reveal whether this new class of mTOR inhibitor would have a desirable efficacy with less toxicity.

### 3.7. Combining mTOR Inhibition with Other Protein Kinase Inhibitors

mTOR responds to both nutrients and growth factors. Other signaling molecules could also be modulated by these signals. Furthermore, the genes encoding the proteins from these signaling pathways also undergo mutations and could drive oncogenesis. Therefore, there are numerous efforts to develop inhibitors to these molecules. One of the signaling pathways that often become deregulated in cancer is the Ras/MAPK pathway. This pathway cross-talks with the PI3K/mTOR pathway and can become highly upregulated to compensate for dampened PI3K/mTOR signals. Indeed, the Ras/MAPK and PI3K/mTOR pathways converge on regulating some common transcriptional regulators of cell growth, proliferation and metabolism [243]. Furthermore, both pathways could also be simultaneously upregulated. In kidney and endometrial carcinoma, mutant Rheb-Y35N led to constitutive activation of both mTORC1 and MEK/ERK pathway, leading to rapamycin resistance of these tumors. Hence, combined mTOR and MAPK inhibition could more effectively prevent growth of tumors that display alterations in these two critical growth-regulatory pathways [294].

The efficacy of combining mTOR inhibition with blockade of the Ras/MAPK pathway to promote cell death is supported by several pre-clinical studies. Combining rapamycin with Raf inhibitors in melanoma cell lines led to more potent growth inhibition [295,296]. The combination of PI3K/mTOR inhibitor CMG002 with sorafenib inhibited proliferation of hepatocellular carcinoma cell lines, induced apoptosis and blocked the activity of both Ras/MAPK and PI3K/mTOR pathways [228]. In thyroid cancer cell lines, BEZ235 combined with RAF265, a pan-RAF inhibitor synergistically inhibited growth of cell lines and mouse xenografts [257]. The mTORC1/2 inhibitor AZD8055 was combined with PI3K inhibitor (GDC0941) and MEK1/2 inhibitor selumetinib each at low doses in advanced stage ovarian clear cell carcinoma cell lines and patient-derived xenograft models [258]. The triple combination was more effective in preventing tumor growth and had better tolerability in PDX models. In a subset of non-small cell lung cancer that have increased rictor expression, the concomitant inhibition of mTORC1/2 and MEK1/2 had synergistic anti-tumor effects [297]. Analysis of genomic and expression data from patient samples revealed that the rictor-altered cohort had increased K-ras/MAPK axis mutations. In leukemia cells wherein Akt is not constitutively active, treatment with an inhibitor of MAPK-interacting kinases (Mnks) increased the sensitivity to rapamycin, resulting in more effective inhibition of proliferation [260]. Whereas Mnk inhibition alone was cytostatic and did not fully block 4EBP phosphorylation, combined Mnk and mTORC1 inhibition was cytotoxic and strongly inhibited 4EBP phosphorylation.

Despite the promising pre-clinical evidence, clinical trials using combined mTOR and Ras/MAPK pathway inhibition of different cancer types have had varied results. In a phase I/II study of patients with recurrent glioblastoma, combination treatment with temsirolimus and sorafenib (Raf kinase and VEGFR-2 inhibitor) led to significant toxicity and lacked efficacy [254]. In phase I studies of everolimus and sorafenib for advanced renal cell cancer, there was enhanced antitumor activity and reasonable tolerability [255,256]. In a phase Ib trial of combined everolimus and the oral MEK inhibitor trametinib (GSK1120212) for patients with advanced solid tumors, there was modest activity and poor tolerability [261]. In a phase Ib dose-escalation study of patients with advanced solid tumors treated with MEK inhibitor trametinib in combination with PI3K/mTOR inhibitor GSK2126458, there was poor tolerability and responses were minimal despite upregulation of PI3K/Ras pathway, likely due to toxicities [229]. In advanced solid tumors, a phase Ib clinical trial of combined MEK inhibitor (pimasertib) and PI3K/mTOR inhibitor (voxtalisib) had poor long-term tolerability and limited anti-tumor activity [262]. In a phase I trial of temsirolimus combined with pimasertib for patients with advanced solid tumors, there was unfavorable toxicity profile although some patients had some clinical benefit and stabilized disease [263]. A randomized phase II trial of the MEK inhibitor selumetinib in combination with temsirolimus for soft tissue sarcomas (STS) revealed that the combination treatment compared to selumetinib treatment alone did not improve progression-free survival in patients with advanced STS. However, the combination treatment seems to have better efficacy for leiomyosarcoma, thus warranting further investigation [259]. It is notable that previous studies have reported that over 70% of primary leiomyosarcoma tumors have increased Akt/mTOR signaling [298]. Hence, future studies should address whether better tolerability and enhanced efficacy would be achieved by tailoring the doses of each inhibitor depending on specific alterations in each or both of the PI3K/mTOR and Ras/MAPK pathways.

### 3.8. Combining mTOR Inhibition with Conventional Chemotherapies and Other Targeted Therapies

Traditional cytotoxic chemotherapy remains a staple in cancer treatment. Many of these drugs, such as alkylating agents, intercalating drugs, microtubule disruptors, and topoisomerase inhibitors, directly target the DNA of the cell [299]. Due to eventual development of resistance, these conventional chemotherapeutic agents are combined with more targeted therapy including mTOR inhibitors. Rapalogs are also largely cytostatic and would likely have more efficacy when combined with cytotoxic chemotherapy. In a phase II trial of everolimus and carboplatin, this combination displayed efficacy in treating patients with triple negative metastatic breast cancer [264]. However, the addition of everolimus to carboplatin in patients with metastatic prostate cancer had minimal clinical efficacy [265]. In a phase II clinical trial of everolimus combined with paclitaxel, which blocks the cell cycle by stabilizing the microtubules, and carboplatin as first-line treatment for metastatic large-cell neuroendocrine lung carcinoma, the treatment was well tolerated and displayed efficacy [267]. In a randomized phase II study in patients with triple negative breast cancer, addition of everolimus to the paclitaxel/cisplatin combination was associated with more adverse events without improvement in clinical response compared to paclitaxel/cisplatin treatment [268]. In a phase I/II clinical trial of everolimus combined with gemcitabine/cisplatin for metastatic triple negative breast cancer, the combination treatment did not have synergistic effects despite the majority of patients harboring PIK3CA mutations [269]. The combination of ridaforolimus with paclitaxel and carboplatin in a phase I study in patients with solid tumor cancers showed antineoplastic activity with no unanticipated toxicities [266]. In a randomized phase II study to compare the effects of monotherapy with vinorelbine, which disrupts microtubules, versus combined vinorelbine and everolimus for second-line chemotherapy in advanced HER2-negative breast cancer, the combined therapy was not superior to the monotherapy, although the treatment was well tolerated [270]. In a phase II study of everolimus in combination with CHOP (cyclophosphamide, doxorubicin, vincristine, and prednisone) as a first-line treatment for patients with peripheral T-cell lymphoma, the treatment was efficacious [271]. Immunohistochemistry revealed maintenance of PTEN expression among patients displaying a complete response. In a phase II study of temsirolimus with liposomal doxorubicin for patients with recurrent and refractory bone and soft tissue sarcomas, stable disease was achieved in more than half of the patients and therapy was well tolerated [272]. The response to treatment correlated with a decline in the highly expressing aldehyde dehydrogenase (ALDH) population of putative sarcoma stem cells, thus supporting that mTOR inhibition could sensitize this population to doxorubicin treatment. In relapsed childhood acute lymphoblastic leukemia, everolimus combined with four-drug reinduction chemotherapy (vincristine, prednisone, pegaspargase and doxorubicin) was well tolerated and had promising results [273]. In a phase I/II study of everolimus in combination with hyperCVAD chemotherapy in patients with relapsed/refractory T-ALL, the combination treatment was well tolerated and produced some favorable patient response. Interestingly, patients that had lower baseline 4EBP phosphorylation at Thr37/46 were associated with a better response to the therapy [300]. A phase I trial of temsirolimus and intensive re-induction chemotherapy in children with relapsed ALL also induced remission in about half of the patients, despite resulting in excessive toxicity at the dosages used [301]. In a study that used temsirolimus as maintenance therapy in castration-resistant prostate cancer after docetaxel induction, the regimen proved safe, and delayed the time to treatment failure to 6 months [274]. Voxtalisib plus temozolomide, an alkylating agent, with or without radiotherapy also displayed a favorable safety profile and moderate amount of PI3K/mTOR pathway inhibition in patients with high-grade glioma [275]. Based on these clinical trial results, combining mTOR inhibition with conventional cytotoxic chemotherapy warrants further investigation. These cytotoxic chemotherapies can overcome the cytostatic effects of rapalogs. As these drugs have also been used over decades, there are also better protocols available for the use of these drugs that could serve to prevent unwanted side effects. Whether the combined mTOR inhibition and cytotoxic chemotherapy could delay the development of resistance remains to be further investigated.

Aside from the traditional cytotoxic chemotherapies, a few conventional targeted therapies have also been used in combination with mTOR inhibition. In a randomized phase II trial in post-menopausal women with hormone receptor positive, EGFR2-negative metastatic breast cancer resistant to aromatase inhibitor therapy, the combination of everolimus with fulvestrant, which downregulates the estrogen receptor, enhanced efficacy compared to fulvestrant alone [279]. In a phase III trial, everolimus combined with the aromatase inhibitor exemestane improved the progression free survival of postmenopausal hormone-receptor-positive advanced breast cancer [277]. In a phase II trial, voxtalisib was used in combination with letrozole, an aromatase inhibitor, in patients with HR+, HER2-negative metastatic breast cancer refractory to a non-steroidal aromatase inhibitor. Voxtalisib, in combination with letrozole, had an acceptable safety profile, but no objective response was observed and further investigation is not warranted [278]. These findings also reveal that combining targeted therapies with rapalogs has more efficacy than with PI3K/mTOR inhibitors. The most likely explanation for this is the increased durability of response and the lower toxicity with the use of rapalogs.

### 3.9. Co-targeting mTOR and Metabolism

The increased demand for nutrient-derived metabolites for macromolecule synthesis by cancer cells is a vulnerability that is often exploited for cancer therapy. The use of anti-metabolites or metabolite analogs can block the generation of critical building blocks for protein, DNA/RNA or lipid synthesis as well as production of reducing equivalents such as NADPH, thus promoting cell death. The mTOR pathway is often upregulated in cancers and promotes generation of these critical metabolic intermediates via modulating the expression or activity of metabolic enzymes or transcription factors. Thus, the dependence of cancer cells to a specific metabolic pathway makes such a pathway a viable target for inhibition and could prove to be more effective especially when combined with mTOR inhibition.

mTORC1 plays numerous roles in the pentose phosphate pathway and nucleotide synthesis. Hence, rapalogs have been used in combination with nucleotide/nucleoside analogs (Table 3). For example, 5-fluorouracil, a pyrimidine analog that has been widely used in cancer therapy has been used in combination with mTORC1 inhibition in recent clinical trials. In a phase Ib study of everolimus plus mFOLFOX-6, a combination chemotherapy regimen of 5-FU, folinic acid, and oxaliplatin, 83% of patients with metastatic gastroesophageal adenocarcinoma experienced a partial response [276]. In phase I trials of everolimus combined with capecitabine, which is a pro-drug of 5-FU, the treatment was found to be well tolerated and safe with promising clinical benefit in metastatic triple negative breast cancer and in advanced solid malignancies [302,303]. Gemcitabine is a nucleoside analogue used to treat several types of cancers. In a phase II study, 48.5% of patients with osteosarcoma were observed to have stabilized disease when treated with a combination of gemcitabine and sirolimus [304]. However, in advanced pancreatic cancer, combination treatment with temsirolimus and gemcitabine lacked clinical efficacy [305]. In a phase Ib/II study of everolimus in combination with azacitidine, a cytidine analog used in patients with relapsed/refractory acute myeloid leukemia, the treatment was tolerable and has promising clinical activity [306]. While these recent studies suggest that combined inhibition of mTORC1 and nucleotide metabolism may have efficacy in certain cancers, it may be worth evaluating mTORC1 signaling and its consequences in reprogramming nucleotide metabolism in these tumors. In a recent pre-clinical study, it was found that mTORC1 inhibition may not provide benefit for tumors with increased mTORC1 activity due to TSC deficiency. Since mTORC1 coordinately controls many anabolic processes, blocking mTORC1 activity would then generally dampen metabolic processes, thus having a cytostatic rather than cytotoxic effect. Instead, targeting one anabolic branch, in this case, inhibiting nucleotide synthesis using the IMP dehydrogenase inhibitor mizoribine, creates a metabolic imbalance, thus promoting cell death [307].

Given the role of mTOR in protein synthesis and amino acid metabolism, mTOR inhibition is also often combined with drugs that block amino acid uptake or those that prevent amino acid biosynthesis. For example, pre-clinical studies demonstrated that combined rapamycin and L-asparaginase, which depletes cells of both glutamine and asparagine, can synergistically inhibit the growth of KRAS-mutant colorectal cancer cells that have upregulated asparagine synthetase [64].

Fatty acid metabolism is also being targeted in combination with mTOR inhibitors for cancer therapy. In pre-clinical studies of ER/HER2 positive breast cancer, cerulenin, a FASN inhibitor synergized with rapamycin to induce apoptosis and inhibit tumorigenesis [308]. FASN inhibition in ovarian cancer led to cell death, which involved a caspase 2 mechanism via the mTORC1 negative regulator REDD1 [309].

Another drug that targets metabolism and has been used in combination with mTORC1 inhibition is metformin. Metformin is a first-line anti-diabetic drug that inhibits the mitochondrial respiratory chain (complex I) and activates the enzyme AMP-activated protein kinase (AMPK). Through its action on AMPK, metformin is believed to also inhibit mTOR as well as activate the tumor suppressor gene TSC2. Two different phase I clinical trials have combined temsirolimus with metformin. In one study, 56% of patients with advanced or refractory cancers were observed with stable disease after treatment of temsirolimus and metformin [310]. In another study, one of 11 patients experienced partial response while five of the remaining patients experienced stable disease. One of these patients with melanoma had stable disease for 22 months [311]. Combined everolimus and metformin in advanced solid malignancies was poorly tolerated likely due to pharmacointeractions between the two drugs since everolimus delayed and prevented the elimination of metformin [312]. In a pharmacodynamic study, patients with advanced solid tumors treated with combined sirolimus and metformin tolerated the regimen but no significant changes in mTOR inhibition or other serum pharmacodynamic biomarkers were observed [313]. The presence of predictive biomarkers of metabolism could improve the efficacy of combined metformin and mTOR inhibitor treatment. For example, in diffuse large B cell lymphoma (DLBCL) wherein about 40% are refractory to the standard combined immunotherapy (R-CHOP), the low levels of GAPDH, which predict poor response to R-CHOP, correlated with dependence on oxidative phosphorylation (OxPhos) metabolism, mTORC1 signaling and glutaminolysis. Based on such dependence, three out of four patients treated with asparaginase, temsirolimus and metformin (KTM) displayed a complete response to this combination treatment [314].

mTORC1 negatively regulates catabolic processes such as autophagy. Hence, inhibition of mTORC1 unleashes autophagy and provides salvaged metabolites to cancer cells thereby allowing their growth and proliferation. Pre-clinical studies combining mTOR and autophagy inhibition displayed synergistic cytotoxic effects [315]. In a phase I clinical trial, significant anti-tumor activity was observed in melanoma patients treated with both temsirolimus and hydroxychloroquine, an autophagy inhibitor [316]. In a phase I/II trial of everolimus with hydroxychloroquine in patients with previously treated renal cell carcinoma, the combination treatment was tolerable and achieved a >40% 6-month PFS rate [317].

As discussed in earlier sections, our knowledge of the roles of both mTOR complexes in controlling metabolic pathways is expanding. As we gain understanding of the unique metabolic needs of different tissues and how they become deregulated in cancer, we can tailor treatment strategies based on their metabolic dependencies. An intriguing concept is the integration of dietary manipulation with targeted therapies to improve patient response [330].

### 3.10. Other Inhibitors of the mTOR Pathway

Compounds that target key signaling molecules along the mTOR pathway have also been developed and are undergoing pre-clinical and clinical trials. A small molecule inhibitor of Rheb, NR1, binds Rheb in its switch II domain and selectively blocks mTORC1 signaling but not mTORC2 or ERK signaling in multiple cell lines [318]. The farnesyl transferase inhibitor lonafarnib, which inhibits the farnesylation of proteins, including Ras and Rheb, downregulates mTOR signaling and potentiates the apoptotic effect of the pan-Raf inhibitor sorafenib but not the Akt inhibitor on melanoma cells [319]. A phase I trial of temsirolimus with the Akt inhibitor, perifosine, for recurrent pediatric solid tumors including gliomas and medulloblastomas, was well tolerated, although partial or complete responses were not achieved [320]. It is not clear if the drugs reached their targets such as the brain. In a phase I study of temsirolimus with perifosine, the combination was generally tolerated with no dose-limiting toxicity and safe in patients with recurrent/refractory pediatric solid tumors [320]. A phase I trial of combined ridaforolimus and an Akt inhibitor (MK-2206) in patients with advanced malignancies showed promising activity in hormone-positive and -negative breast cancer with PI3K pathway dependence [321]. In a large panel of cancer cell lines, T-cell acute lymphoblastic leukemia (T-ALL) with Notch mutation were highly sensitive to Akt inhibitor (AZD5363) and the mTORC1/2 inhibitor (AZD2014) but only partially sensitive to PI3K inhibitors [325]. While combining mTOR with Akt inhibition showed promise with less toxicity, it seems that Akt inhibitor monotherapy had low clinical activity partly due to poor tolerability [322,323,324]. Pre-clinical studies on inhibition of SGK, another target of mTORC2, for the treatment of a variety of cancers are also showing promise [327,328,331]. SGK1 could play a role in enhancing uptake of unsaturated fatty acids in hypoxic lung cancer cell [327]. The hypoxia makes these tumors reliant on uptake rather than desaturation of saturated FA, a process that occurs in normoxia. Therefore, this vulnerability can also be exploited for more effective therapy of tumors that have deregulated fatty acid metabolism. The 3-phosphoinositide-dependent kinase 1 (PDK1) inhibitor GSK2334470 displayed antitumor activity in multiple myeloma cells. Interestingly, while it blocked phosphorylation of the mTORC1 target S6K1 (at Thr389) and the phosphorylation of Akt at Thr308, it did not affect phosphorylation of Akt at the mTORC2-targeted site Ser473. Combined treatment with GSK2334470 and the mTOR inhibitor PP242 had more potent anti-myeloma activity and led to complete inhibition of mTOR and Akt [329]. Future studies should reveal which types of tumors could benefit from this combination treatment in the clinic.

### 3.11. Resistance Mechanisms and Other Therapeutic Opportunities

A main challenge in cancer therapy is the development of resistance to chemotherapeutic agents and targeted therapies by malignant cells. As for mTOR inhibitors, cells acquire mutations in mTOR or its partners that could prevent drug binding or upregulate mTORC components that could enhance mTOR activity, hence limiting the efficacy of these inhibitors [231,332]. Cells can also bypass a block in mTOR by upregulating other signaling and metabolic pathways. Non-biased omics technologies have facilitated identification of such bypass mechanisms that could serve as therapeutic opportunities or predictive biomarkers for follow-up treatment. Recent efforts to gain insights into such resistance mechanisms underscore the ability of cancer cells to restore key metabolic processes that support their growth and proliferation. For example, proteomics and genomics studies that analyzed changes that occur after treatment with mTOR inhibitors revealed bypass mechanisms related to protein synthesis in Ewing sarcoma cells [333] and glioblastoma [334]. Using metabolomics studies, another resistance mechanism occurring due to PI3K/mTOR inhibition is via upregulation of the purine salvage pathway in small cell lung carcinoma [335]. Non-genetic mechanisms can also occur, as has been shown recently in a single-cell phosphoproteomics analysis of patient-derived in vivo glioblastoma model. Resistance to TORKI was monitored by single cell phosphoproteomics and revealed adaptive signaling dynamic alterations that were responsive to combinations of drugs targeting these pathways [336]. The combination of genomics, proteomics and metabolomics and improved technologies in single-cell analysis should have a tremendous impact in moving towards more personalized therapy.

### 3.12. Immunotherapy

mTOR plays a central role in immunity [337]. In fact, rapalogs are widely used as immunosuppressants to prevent kidney transplant rejection. Interestingly, rapamycin affects only distinct classes of immune cells [338]. Studies on the metabolic dependencies of specific T cell subsets are providing clues on the basis of rapamycin’s specific effects [339]. Quiescent or naïve T cells rely on OxPhos for their metabolic needs, whereas activated highly proliferating effector T cells depend on robust glycolytic metabolism to fuel their growth and proliferation [340,341,342]. Inhibition of mTOR using rapalogs specifically affects these highly proliferating immune cells while being ineffective on T cells that rely on OXPHOS metabolism, such as T-regulatory (T-reg) and memory T cells. The effects of mTOR inhibition on specific T-cell subsets can be exploited to improve cancer treatment and immunotherapy. T cells are an important component of the tumor microenvironment. Inhibitors that promote cancer cell death combined with strategies to boost effector functions of T cells while repressing negative regulators of the immune responses could improve therapeutic outcome. In a phase I clinical trial for the treatment of metastatic renal cell cancer, everolimus was combined with low-dose cyclophosphamide [343]. Cyclophosphamide was administered to selectively deplete the immunosuppressive T-regs, which undergo expansion in the presence of rapamycin [344]. The results have been promising as the treatment sustained levels of CD8^+^ T-cell population together with increased effector to suppressor ratio. The observed changes in various immune cell populations may promote antitumor immunity. These promising results have led to a phase II clinical trial. mTOR inhibition in the tumor may also have consequences that allow T cells to enhance their anti-tumor recognition. In pre-clinical studies, everolimus combined with anti-PD-L1 was more effective in decreasing tumor burden compared to individual treatment in a mouse model of renal cell carcinoma due to upregulation of PD-L1 in the tumor cells resulting in increased tumor infiltrating CD8+ lymphocytes and tumor regression [345]. Manipulating mTOR signaling in other immune cell types is also being investigated for enhancing immunotherapeutic strategies. Inhibition of mTOR using rapamycin increases cytotoxicity of Vγ4 γδ T cells towards various cancer cell lines by enhancing NKG2D expression and TNF-α expression [346].

Immunotherapeutic approaches using ex vivo cultures for adoptive transfer of genetically modified T-cells including tumor-infiltrating lymphocytes and chimeric antigen receptor-T cells (CAR-T) could also be enhanced by manipulation of mTOR activity. Akt inhibition during ex vivo expansion of tumor-infiltrating lymphocytes increased the generation of antitumor CD8+ T cells with memory cell phenotypes. This allows increased persistence following adoptive transfer, thus enhancing their antitumor activity [347]. Decreased mTORC1 activity due to IL15, which is used to expand CAR-T cells, also enhances CAR-T cell antitumor activity by preserving their stem cell memory phenotype. CAR-T cells that are less differentiated or less exhausted are more effective [348].

mTOR inhibition is also being exploited for improvement of vaccine strategies. Rapalogs can enhance the generation of CD8^+^ memory T cells in response to vaccination. This appears to be due to the rapamycin-mediated reprogramming of metabolism to fatty acid oxidation [349,350]. T cells that have Rheb-deficiency, thereby decreased mTORC1 activity, have poor effector cells while their memory cells persisted. Conversely, T cells that are TSC2 null, thereby with hyperactive mTORC1, have increased effector phenotypes but fail to convert to memory cells [351]. mTOR inhibition also improved CD8^+^ T cell responses to vaccinia virus vaccination in rhesus macaques [352] and enhanced immune responses to influenza vaccine in the elderly [353]. Using microparticles encapsulating rapamycin, the low dose release of rapamycin polarized vaccine-induced T cells toward central memory phenotypes, which could enhance anti-tumor immunotherapy [354]. Rapalogs could also modulate dendritic cell function thus enhancing anti-tumor effects of DNA vaccines [355,356]. A better understanding of the metabolic dependencies of various immune cells should provide more opportunities for immunotherapy.

## 4. Conclusions

Less than a century after Warburg’s hypothesis, we now have a better understanding of how the defective metabolic processes in cancer cells are intertwined with genetic and proteomic alterations. The discovery of rapamycin and the mTOR pathway facilitated the elucidation of how cancer cells reprogram their metabolism in order to acquire nutrients that are necessary for their growth and proliferation. Perturbations in this pathway, such as oncogenic and tumor suppressor mutations that elevate mTOR signaling, lead to rewiring of metabolic pathways in ways that increase aerobic glycolysis, as Warburg reported. However, studies over the past decades unravel that cancer cells also display heterogeneous metabolic vulnerabilities that can be exploited for more effective and specific therapy. The results from rapalog clinical trials suggest how tumors that may have particular metabolic signatures due to mTORC1 activation, by virtue of mutations in TSC, are more sensitive to rapalog monotherapy. There are numerous efforts to identify predictive biomarkers and thus identify patients who would benefit most from mTOR inhibitors [239,357,358,359]. Such biomarkers include not only genetic, proteomic or signaling alterations but also metabolite changes. The latter is exemplified by recent studies in gliomas harboring the mutant isocitrate dehydrogenase 1 that produces the oncometabolite 2-hydroxyglutarate. These gliomas displayed sensitivity to voxtalisib, the dual PI3K/mTOR inhibitor [360]. Although other metabolites were altered upon voxtalisib treatment, the decrease in 2HG highly correlated with the increased animal survival. Thus, measurement of 2HG by magnetic resonance spectroscopy could be useful as a metabolic biomarker for mutations in isocitrate dehydrogenase 1 (IDHmut) glioma response to PI3K/mTOR inhibition. The mTOR pathway can also be activated by multiple mechanisms. In a pan-cancer proteogenomic analysis of thousands of human cancers, many of these cancers had high mTOR pathway activity despite the lack of alterations in canonical genes associated with this pathway [361]. These findings highlight the need to explore metabolomic impacts on mTOR signaling. In other studies, despite rationalizing treatment strategy based on molecular aberrations in the PI3K/mTOR and EGFR/MAPK pathways, patients did not derive significant benefits [362]. Such resistance to mTOR inhibition underscores how the metabolic plasticity of cancer cells enables the emergence of alternative mechanisms to feed the growing tumor. Other signaling pathways that converge on mTOR or metabolism could also serve as potential additional targets. For example, CDK4, which regulates the cell cycle, modulates mTORC1 via phosphorylation of the tumor suppressor FLCN [363]. This regulates mTORC1 recruitment to the lysosomal surface in response to amino acids. Epigenetic mechanisms could also contribute to metabolic reprogramming. In a phase I study using combined ridaforolimus and the histone deacetylase (HDAC) inhibitor vorinostat in advanced renal cell carcinoma, prolonged disease stabilization was observed and was tolerable at the phase II dose [364]. In a phase I study of sirolimus and vorinostat in patients with advanced malignancy, the combination treatment seemed to be safe and displayed efficacy [189]. Identification of mutations in mTOR could also inform on sensitivity to mTOR inhibitors. Indeed, mTOR mutations have been identified in human cancers [163,164,365,366,367,368,369]. Toxicities associated with mTOR inhibitors could also be addressed by the mode of drug delivery. Improvement of formulation and dosing could be beneficial. A twice daily 5 mg dosing instead of once-daily 10 mg regimen conducted on a randomized pharmacokinetic crossover trial of everolimus suggested that such dosing could be promising to reduce adverse toxicity while maintaining treatment efficacy [370]. Nanoparticle-based mTOR targeting would need to be improved as well in order to have more specific effects on tumors and prevent undesirable cellular perturbations [371]. Defining the metabolic conditions within a given tumor microenvironment would also improve targeting strategies and perhaps provide clues on metastatic sites that could be conducive for growth of a malignant tumor with a metabolic dependency. Furthermore, given the dynamic effects of nutrition in metabolism, gene expression and cell signaling between different tissues and individuals, a deeper understanding of the mechanisms involved in these processes should pave the way for integrating diet manipulation with targeted therapeutic strategies and more personalized therapy.

An exciting application of mTOR inhibitors is in the improvement of immunotherapeutic strategies. The use of mTOR inhibitors for the improvement of vaccination strategies also promises to have a major impact towards cancer vaccination. As we gain more understanding of the metabolic needs of different immune cell subsets as well as the metabolic vulnerabilities of the tumor in the TME, we can improve anti-tumor activity of effector T cells while preventing immunosuppressive mechanisms as well as develop more innovative targeted therapeutic strategies for more effective cancer treatment.

## Figures and Tables

**Figure 1 cells-08-01584-f001:**
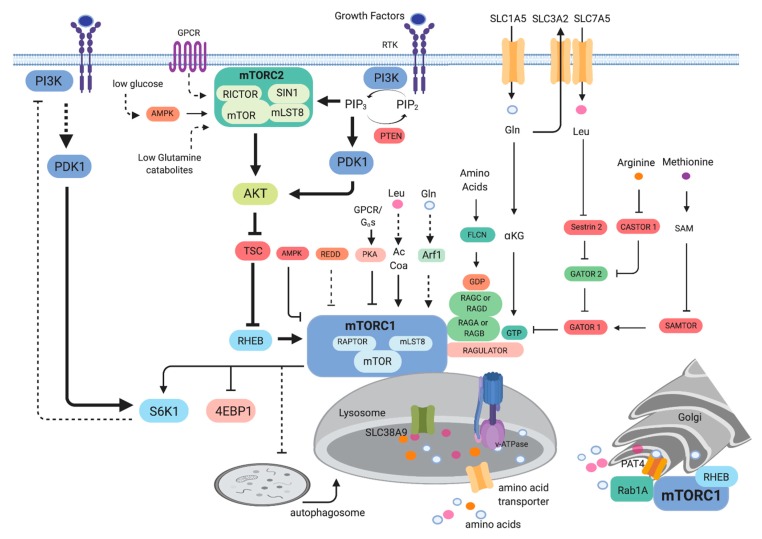
mTOR Signaling. mTORC1 activation is modulated by the presence of nutrients such as amino acids at the membrane surface of organelles such as the lysosomes and Golgi. Signaling to mTORC1 is potentiated by growth factor/PI3K signaling via Akt. mTORC2 activation is enhanced by the presence of growth factors and also occurs on membrane subcellular compartments. It is also augmented by G-protein coupled receptor (GPCR) signaling and by nutrient-limiting conditions. The bold lines indicate signals from growth factor signaling. The dashed lines indicate indirect modulation.

**Figure 2 cells-08-01584-f002:**
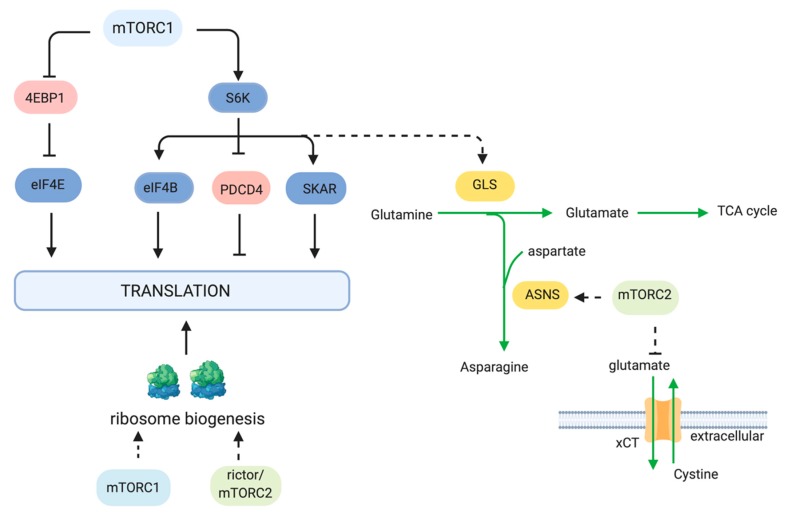
The role of mTOR in protein synthesis. mTOR controls several aspects of protein synthesis, including ribosome biogenesis, translation, and amino acid transport and synthesis.

**Figure 3 cells-08-01584-f003:**
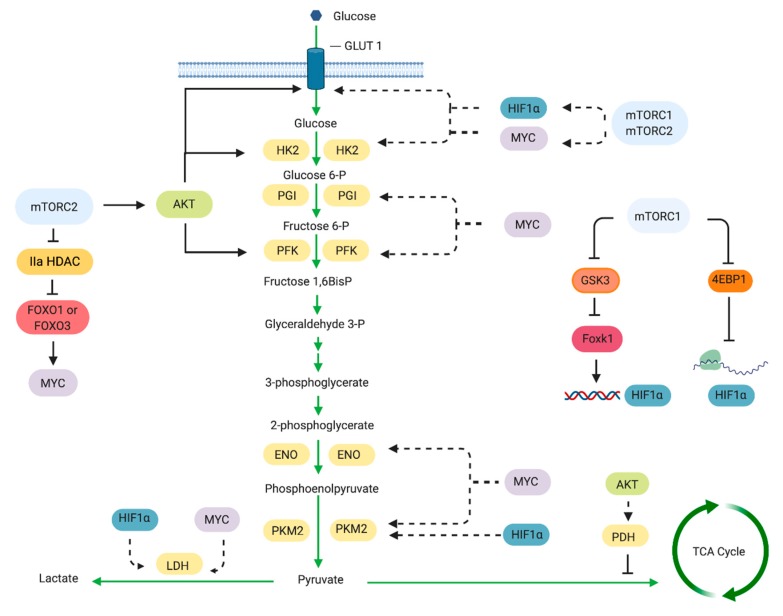
The role of mTOR in glucose metabolism. mTORC1 controls glycolysis via HIF1α and MYC. Through these transcription factors, it promotes the expression of genes whose products are involved in glucose metabolism including glucose transporters and glycolytic enzymes. mTORC2 also has a positive role in the regulation of glucose metabolism. The mTORC2 target, Akt, modulates glucose transport and phosphorylates glycolytic enzymes.

**Figure 4 cells-08-01584-f004:**
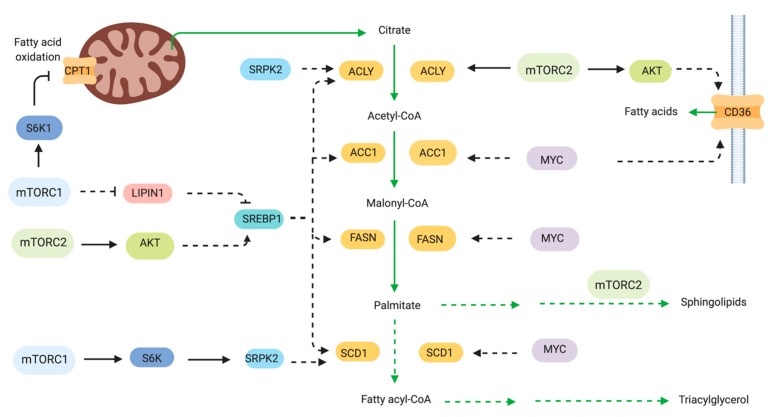
The role of mTOR in fatty acid and lipid synthesis. mTOR controls metabolic enzymes involved in fatty acid and lipid synthesis via SREBP1 and SRPK2. They also modulate expression of fatty acid transporters such as CD36 and carnitine palmitoyl transferase 1c (CPT1c).

**Figure 5 cells-08-01584-f005:**
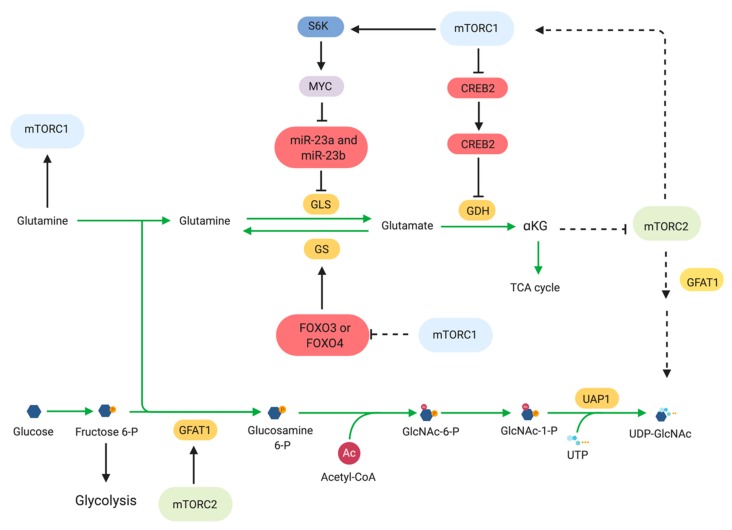
The role of mTOR in glutamine metabolism and hexosamine biosynthesis. mTORC1 promotes glutaminolysis and is active in the presence of sufficient glutamine. On the other hand, mTORC2 activation is enhanced when glutamine catabolite levels diminish to promote flux through the hexosamine biosynthesis pathway via control of GFAT1.

**Figure 6 cells-08-01584-f006:**
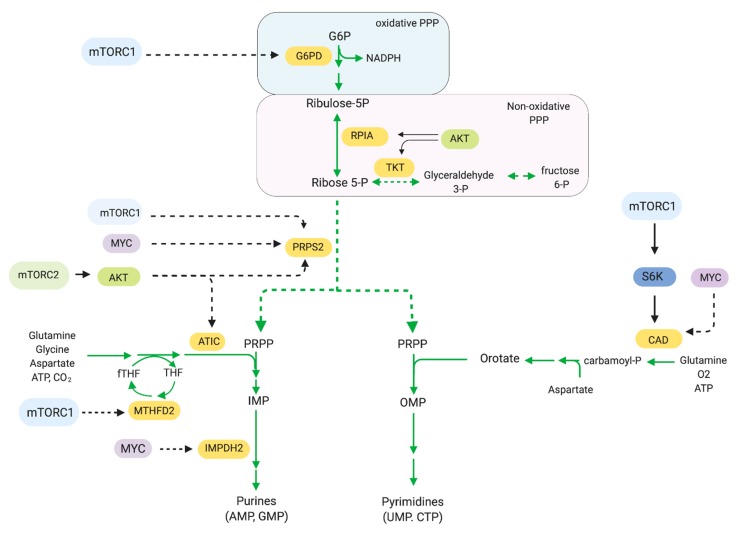
The role of mTOR in the pentose phosphate pathway and nucleotide synthesis. mTOR modulates the expression of the enzymes involved in the pentose phosphate pathway, purine and pyrimidine synthesis.

**Table 1 cells-08-01584-t001:** mTOR or mTOR complex inhibitors that are currently used in pre-clinical and clinical studies.

Drug	Target	References
**Rapalogs (Rapamycin Analogs)**		
Sirolimus	mTORC1/FKBP12	[188,189]
Everolimus (Affinitor, RAD001)	mTORC1/FKBP12	[171,172,173,174,175,176,177,181,190,191,192,193,194]
Temsirolimus (Torisel, CCI779)	mTORC1/FKBP12	[178,179,180,195]
Ridaforolimus (deforolimus, AP23573)	mTORC1/FKBP12	[182,183,184,185,186,187]
**mTOR ATP-competitive inhibitors (TORKIs)**		
AZD8055	mTOR	[196,197,198,199]
AZD2014, vistusertib	mTOR	[200,201,202,203]
TAK-228; INK-128; sapanisertib	mTOR	[204,205]
OSI-027; ASP7486	mTOR	[206]
MLN0128	mTOR	[207]
CC-223	mTOR	[208]
**Dual PI3K/mTOR inhibitors**		
BEZ235; dactolisib	PI3K/mTOR	[209,210,211,212,213,214,215,216,217,218,219,220]
PKI-587; gedatolisib	PI3K/mTOR	[221,222]
GDC-0980, apitolisib	PI3K/mTOR	[223,224,225]
SAR245409, XL765; voxtalisib	PI3K/mTOR	[226]
BGT226	PI3K/mTOR	[227]
CMG002	PI3K/mTOR	[228]
GSK2126458	PI3K/mTOR	[229]
**Other**		
JR-AB2-011	Rictor	[230]
RapaLink1	mTOR/FKBP12	[231,232,233]

**Table 2 cells-08-01584-t002:** Other targeted or cytotoxic chemotherapy that are used in combination with mTOR inhibitors.

Drug	Target	References
**RTK inhibitors**		
Trastuzumab	HER2	[244,245]
Neratinib	HER2	[246]
Tivantinib	c-Met	[247]
Cetuximab	EGFR	[248,249]
Bevacizumab	VEGF	[249,250]
Dalotuzumab	IGF-1R	[251]
Cixutumumab	IGF-1R	[252]
Pazopanib	Tyrosine Kinase	[164]
Lapatinib	HER2/EGFR	[253]
**Ras/MAPK Pathway Inhibitors**		
Sorafenib	Raf/VEGFR	[228,254,255,256]
RAF265	Pan-Raf	[257]
Selumetinib	MEK1/2	[258,259]
MNKI-57; MNKI-4	Mnk1/2, Mnk2	[260]
Trametinib (GSK1120212)	MEK	[229,261]
Pimasertib	MEK	[262,263]
**Cytotoxic Chemotherapy**		
Carboplatin	DNA	[264,265,266]
Paclitaxel	Microtubules	[266,267,268]
Cisplatin	DNA	[268,269]
Vinorelbine	Microtubules	[244,270]
Cyclophosphamide	DNA	[271]
Doxorubicin	Topoisomerase	[272]
Vincristine	Microtubules	[273]
Docetaxel	Microtubules	[274]
Temozolomide	DNA	[275]
Oxaliplatin	DNA	[276]
**Other Targeted Therapies**		
Exemestane	Aromatase	[277]
Letrozole	Aromatase	[278]
Fulvestrant	Estrogen Receptor	[279]

**Table 3 cells-08-01584-t003:** Metabolism inhibitors or anti-metabolites and other mTOR pathway inhibitors.

Drug	Target	References
**Metabolism Inhibitors**		
5-Fluorouracil	Thymidylate Synthase	[276]
Capecitabine	Thymidylate Synthase	[302,303]
Gemcitabine	DNA	[304,305]
Azacitidine	DNA methyltransferase/DNA	[306]
Mizoribine	IMP dehydrogenase	[307]
L-asparaginase	Asparagine	[64]
Cerulenin	FASN	[308,309]
Metformin	Complex 1	[310,311,312,313,314]
Hydroxychloroquine	Lysosomes/Autophagy	[315,316,317]
**Other mTOR Pathway Inhibitors**		
NR1	Rheb	[318]
Lonafarnib	Farnesyl transferase	[319]
Perifosine	Akt	[320]
MK-2206	Akt	[321,322,323,324]
AZD5363	Akt	[325]
GSK650394	SGK1	[326,327]
SI113	SGK1	[328]
GSK2334470	PDK1	[329]

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
