# Peer review of "Targeting mTOR and Metabolism in Cancer: Lessons and Innovations"

_cells, 2019, doi:10.3390/cells8121584_

Round 1
Reviewer 1 Report
The authors have made a very comprehensive review on targeting mTOR and metabolism in cancer. It is a highly useful one for many readers.
The manuscript is well organized and well written. I do not have much suggestion for revision.
Just two very minor points.
Figure 1. In the legend, it was stated that “signaling to mTORC1 is potentiated by growth factor/PI3K signaling”. However, in the figure, the connection of the two is not so clear. Please make some minor modifications. Tables 1-3. There are two columns only: drug and target. Do you think that it is worth to add some key references as the third column? It is totally up to the authors.Author Response
We thank the reviewer for his/her enthusiasm and helpful comments. We have re-done Figure 1 and modified the Figure 1 legend in order to clarify activation of mTORC1 (see bold lines in figure to indicate growth factor signaling. We also indicated this in the Legend). Since we added some points on the discussion on the main text as requested by Reviewer 2, we also further expanded this Figure.
We have also added the References to the Tables as suggested.
Reviewer 2 Report
This is a thoughtful review suggesting a rationale for the limited success of mTOR inhibitors in the clinic despite prothe promise of pre-clinical studies. I diid have some suggestions for the authors relating to Figure 1.
The authors have included a number of cytosolic amino acid sensors, but have omitted members of the SLC36 and SLC38 amino acid transporter family which have also been highlighted as amino acid sensors. To provide a rounded overview I think it is important that SLC36A1 and SLC38A9 are added to Figure 1 and some discussion of their roles in amino acid sensing and mTORC1 activation added to the text and inclusion of a balanced review on mTORC1 activation by amino acid transporters, such as Goberdhan et al., 2016 cited. Recents studies highlighting different functional pools of mTORC1 would also be helpful to cite and discuss, eg Fan et al., 2016; Morotti et al., 2019; Hateyama et al., 2019.Author Response
We thank the reviewer for his/her positive comments and for the great suggestion to add a discussion on other mechanisms of mTORC1 activation by amino acid sensors/transporters. We have now expanded the discussion on the main body of the text including suggested references (see pages 5-6) and also added these mechanisms on Figure 1.
Reviewer 3 Report
Very comprehensive work!
I have my concerns on the Immunotherapy part.
The authors emphasize the potentiality to manipulate the mTOR pathway to improve immunotherapy via the possible mechanisms,
Attenuation of mTORC1 signals favours expansion of CD8+ memory and CD4+ regulatory T cells. Akt inhibition allows to increase the generation of memory phenotype of antitumor CD8+ T cells . Decreased mTORC1 activity due to IL15, which is used to expand CAR-T cells, help to preserve their stemness.But we already know that Rapalogs are used as immunosuppressant, which inhibits activation of T cells and B cells probably by blockade of IL-2 and IL-15 induction via inhibition of S6K and prevents progression of the cell cycle from the G1 to S phase.
There seems to be a contradiction between its immunosuppressive and immune boost effect. Can more explanation of this part be given?
Author Response
We thank the reviewer for his/her helpful comments. We have now clarified the points raised by the reviewer on the immunotherapy aspects of manipulation of mTOR pathway. We hope that the revised discussion now reflects the distinction between the immunosuppressive vs immunotherapeutic effects of mTOR inhibition (Please see Immunotherapy section on page 29).